# Host-microbe multiomic profiling identifies distinct COVID-19 immune dysregulation in solid organ transplant recipients

Harry Pickering[1,30], Joanna Schaenman[1,30], Hoang Van Phan[2,30], Cole Maguire [3,30], Alexandra Tsitsiklis[2], Nadine Rouphael [4], Nelson Iván Agudelo Higuita[5], Mark A. Atkinson[6], Scott Brakenridge[6], Monica Fung[2], William Messer [7], IMPACC Network*, Ramin Salehi-rad[1], Matthew C. Altman [8], Patrice M. Becker [9], Steven E. Bosinger [4], Walter Eckalbar[2], Annmarie Hoch [10], Naresh Doni Jayavelu[8], Seunghee Kim-Schulze[11], Meagan Jenkins [1], Steven H. Kleinstein [12], Florian Krammer [11], Holden T. Maecker [13], Al Ozonoff [10,14,15], Joann Diray-Arce [10,15], Albert Shaw[12], Lindsey Baden [15,16], Ofer Levy [10,14,15], Elaine F. Reed[1,31] & Charles R. Langelier [2,17,31] ✉

Coronavirus disease 2019 (COVID-19) poses significant risks for solid organ transplant recipients, who have atypical but poorly characterized immune responses to infection. We aim to understand the host immunologic and microbial features of COVID-19 in transplant recipients by leveraging a prospective multicenter cohort of 86 transplant recipients age- and sex-matched with 172 non-transplant controls. We find that transplant recipients have higher nasal SARS-CoV-2 viral abundance and impaired viral clearance, and lower anti-spike IgG levels. In addition, transplant recipients exhibit decreased plasmablasts and transitional B cells, and increased senescent T cells. Blood and nasal transcriptional profiling demonstrate unexpected upregulation of innate immune signaling pathways and increased levels of several proinflammatory serum chemokines. Severe disease in transplant recipients, however, is characterized by a less robust induction of pro-inflammatory genes and chemokines. Together, our study reveals distinct immune features and altered viral dynamics in solid organ transplant recipients.

Coronavirus disease 2019 (COVID-19) has resulted in an enormous societal burden, with a toll of millions of infections and deaths worldwide[1]. Immunocompromised patients who have undergone solid organ transplantation (SOT) are more susceptible to SARS-CoV-2 infection and produce less robust antibody responses following vaccination[2], although they can achieve effective T-cell responses with multiple vaccinations[3]. In addition, they are more likely to be hospitalized, experience adverse clinical outcomes, and have longer durations of infectiousness compared to the general population[4–8]. Despite this, propensity-matched studies demonstrate that COVID-19 mortality in SOT recipients is not higher compared to immunocompetent individuals[5,8–10], although some studies using other matching approaches have found differences[6,11].

To prevent organ rejection, SOT depends on immune suppression with a battery of agents including calcineurin inhibitors (e.g., cyclosporin, tacrolimus), cell cycle inhibitors (e.g., mycophenolate mofetil),

A full list of affiliations appears at the end of the paper. *A list of authors and their affiliations appears at the end of the paper.
✉e-mail: chaz.langelier@ucsf.edu

and corticosteroids. This leads to an altered immunological landscape in SOT recipients, resulting in differing host responses to severe infections, including from SARS-CoV-2. Few studies, however, have profiled the immune landscape of SOT recipients in the context of severe infection[12–14], and none have yet used a multiomic approach to assess their responses at the cellular, protein, transcriptional, and microbial levels.

The distinct immune responses of SOT recipients could theoretically be both detrimental and advantageous in the context of COVID-19. On the one hand, impaired innate and adaptive immunity in SOT recipients increases susceptibility to infection and impairs viral clearance[8], which could lead to worse outcomes. On the other hand, because severe COVID-19 is characterized by a dysregulated, over-exuberant inflammatory response[15–17], the inherent immunosuppression of SOT recipients could confer protection against severe disease. Developing a better mechanistic understanding of this tenuous immune balance in SOT recipients could inform more effective treatment approaches for COVID-19 or other respiratory viral infections, in particular the optimal use of immune modulating therapies[8].

Here, we leverage a multicenter cohort[17–19] of 1164 vaccine-naïve patients hospitalized for COVID-19 to carry out a comprehensive immunoprofiling analysis of host and microbe in SOT recipients with acute SARS-CoV-2 infection. This cohort afforded the opportunity to study immune responses over the course of hospitalization through concurrent analysis of transcriptional, proteomic, cellular, and antibody responses in addition to viral abundance and the airway microbiome. Contrary to expectations, we find that SOT recipients demonstrate a globally heightened innate inflammatory response compared to non-SOT controls, and observe that established biomarkers of COVID-19 severity do not correlate with disease trajectory in this vulnerable demographic.

## Results
### Patient cohort
We conducted a case-control study of patients hospitalized for COVID-19 within the IMMuno Phenotyping Assessment in a COVID-19 Cohort (IMPACC), which comprised 1164 patients enrolled across the US[17–19]

between May 2020 and March 2021. 86 SOT recipients from 11 medical centers were matched 2:1 by age, sex, and study site with 172 non-SOT controls from the same cohort (Fig. 1 and Table 1). The most common transplanted organ type was kidney (Supplementary Table 1), with approximately equal representation of heart, liver, and lung. Immunosuppressive regimens being taken at the time of hospital admission varied across SOT recipients, although mycophenolate and tacrolimus were the most common (Supplementary Table 2). We found no differences between SOT recipients and non-SOT controls in terms of ICU admission, intubation status, or COVID-19 severity as measured by five established COVID-19 outcome trajectory groups (TG)[18], or as measured by 28-day mortality. Trajectory groups (TG) 1–3 had mild to moderate disease based on hospital stay and level of respiratory support, while TG 4 was characterized by longer hospitalizations and prolonged respiratory support requirements, and TG 5 by death within 28 days[18]. Within the SOT group, we asked whether receipt of either mycophenolate or tacrolimus at the time of hospital admission influenced the severity of TG but found no significant differences. Of patients receiving mycophenolate, 37.7% were in TG 4–5, compared with 18.8% of those not receiving mycophenolate ($P = 0.077$). Of patients receiving tacrolimus, 30.4% were in TG 4–5, compared with 28.6% of those not receiving tacrolimus ($P = 0.88$).

To investigate host immunologic and microbial features associated with COVID-19 in SOT recipients, we assessed data from mass cytometry (CyTOF), transcriptional profiling, proteomics, and serologic analyses in the blood, as well as nasal swab transcriptional profiling and metatranscriptomics following hospital admission, and longitudinally at up to six timepoints up to ~28 days post-hospital admission (Fig. 1).

### SOT is associated with increased SARS-CoV-2 viral abundance, and impaired viral clearance
We began our analyses by examining the SARS-CoV-2 viral abundance, as measured in reads per million (rpM) by nasal metatranscriptomic RNA sequencing and N-gene reverse transcription PCR (Supplementary Fig. 1). SOT recipients had significantly higher SARS-CoV-2 viral rpM at Visit 1 ($P = 6.8e-9$, Fig. 2a), which could not be explained by

**Fig. 1 | Study overview.** This study evaluated solid organ transplant recipients ($N = 86$) matched 2:1 with non-transplant controls ($N = 172$) enrolled in the IMPACC cohort of patients hospitalized for COVID-19 at 20 medical sites across the United States. Blood (PBMCs and serum) and nasal swab samples were collected at up to 6 visits over 28 days, and processed for RNA sequencing, proximity extension assay (Olink) soluble proteomics, mass cytometry, and serology. Created in BioRender[32].

## Table 1 | Clinical and demographic features of cohort

|  | SOT cases | Non-SOT controls | P value |
|---|---|---|---|
| **Median age (IQR)** | 57.5 (51.3–64.0) | 58.0 (50.8–63.3) | 0.644 |
| **Site (%)** |  |  | 0.189 |
| Boston/BWH | 7 (8.1%) | 18 (10.2%) |  |
| Case Western | 2 (2.3%) | 7 (4.0%) |  |
| Emory | 5 (5.8%) | 12 (6.8%) |  |
| Florida | 3 (3.5%) | 4 (2.3%) |  |
| ISMMS (Mt Sinai) | 1 (1.2%) | 2 (1.1%) |  |
| OHSU (Oregon) | 2 (2.3%) | 3 (1.7%) |  |
| OUHSC (Oklahoma) | 2 (2.3%) | 0 (0.0%) |  |
| Stanford | 2 (2.3%) | 12 (6.8%) |  |
| UCLA | 40 (46.5%) | 54 (30.7%) |  |
| UCSF | 15 (17.4%) | 47 (26.7%) |  |
| Yale | 7 (8.1%) | 17 (9.7%) |  |
| **Female sex (%)** | 25 (29.1%) | 52 (29.5%) | 1.000 |
| **Early enrollment (%)** | 39 (45.3%) | 78 (44.3%) | 0.172 |
| **Ethnicity (%)** |  |  | 0.965 |
| Hispanic or Latino | 49 (57.0%) | 96 (54.5%) |  |
| Not Hispanic or Latino | 36 (41.9%) | 79 (44.9%) |  |
| Not Specified | 1 (1.2%) | 1 (0.6%) |  |
| **Race (%)** |  |  | 0.393 |
| American Indian/Alaska Native | 1 (1.2%) | 1 (0.6%) |  |
| Asian | 2 (2.4%) | 3 (1.7%) |  |
| Black/African American | 15 (17.4%) | 30 (17.0%) |  |
| Multiple | 1 (1.2%) | 0 (0.0%) |  |
| Other/Declined | 34 (39.5%) | 81 (46.0%) |  |
| Unknown/Unavailable | 3 (3.5%) | 2 (1.1%) |  |
| White | 30 (34.9%) | 59 (33.5%) |  |
| **Trajectory group** |  |  | 0.808 |
| 1 | 16 (18.6%) | 35 (20.3%) |  |
| 2 | 20 (23.3%) | 38 (22.1%) |  |
| 3 | 26 (30.2%) | 40 (23.3%) |  |
| 4 | 19 (22.1%) | 52 (30.2%) |  |
| 5 | 5 (5.8%) | 7 (4.1%) |  |
| **ICU admission (%)** | 32 (37.2%) | 64 (37.2%) | 1.00 |
| **Ever intubated (%)** | 17 (19.3%) | 42 (23.4%) | 0.497 |
| **Mortality (%)** |  |  |  |
| D28 | 5 (5.8%) | 7 (4.1%) | 0.754 |
| Ever | 12 (14.0%) | 20 (11.6%) | 0.739 |
| **Diabetes (%)** | 38 (44.2%) | 55 (32.0%) | 0.074 |
| **Steroids (%)** | 76 (88.4%) | 104 (60.5%) | 8.3e-6 |
| **Remdesivir (%)** | 57 (66.3%) | 124 (72.1%) | 0.414 |

differences in the time from symptom onset ($P = 0.16$, Supplementary Fig. 2), and did not differ based on the type of transplanted organ ($P = 0.65$, Fig. 2b, Supplementary Table 1) or receipt of either mycophenolate or tacrolimus (Supplementary Fig. 3a, b). Longitudinal analysis revealed that there was a significant association between SOT status and viral rpM, with SOT recipients demonstrating impaired viral clearance compared to non-SOT controls (Fig. 2c, $P = 0.0022$).

### Immune cell populations and SARS-CoV-2 antibody levels

To measure immune cell populations in blood, we used mass cytometry (CyTOF) with a panel of 43 antibodies designed to identify cell lineages and markers of functional status. In PBMC samples from Visit 1, we found 5 cell types associated with SOT status (false discovery rate (FDR) < 0.05) (Fig. 3a). Plasmablasts and transitional B cells were significantly less abundant at Visit 1 in SOT recipients compared to controls (Fig. 3a, b). Conversely, SOT recipients demonstrated increased proportions of CD4 + T (EMRA CD57hi) and CD4 + T (EMRA CD57low) cells, and CD8 + T (EMRA CD57low) cells (Fig. 3a, b). After adjusting for SARS-CoV-2 viral rpM, only plasmablasts and CD4 + T (EMRA CD57low) cells remained statistically significant in terms of proportional differences, suggesting that these two cell types were associated with SOT status in a viral abundance-independent manner (Supplementary Fig. 4).

We also compared anti-SARS-CoV-2 spike IgG levels between the two groups. SOT recipients had lower antibody levels at Visit 1 ($P = 0.0004$, Fig. 3c), although the rates of increase did not differ based on SOT status (Fig. 3d). Anti-SARS-CoV-2 spike IgG levels also did not differ based on receipt of either mycophenolate or tacrolimus (Supplementary Fig. 3c, d).

### Cytokine and chemokine expression upon hospitalization and over time

Analysis of proximity extension assay (Olink) proteomics data from serum samples identified 18 proteins differentially expressed based on SOT status at Visit 1 (Fig. 4a). The expression levels of 14 of these proteins were higher in SOT recipients versus non-SOT controls, including CX3CL1, IL15RA and KITLG (Fig. 4b). SOT recipients had lower levels of IFN-gamma (IFNG), OSM, TNSF14, and CCL4.

To assess whether differences in SARS-CoV-2 rpM may contribute to the observed differential protein expression, we repeated the analyses with adjustment for viral rpM. We found that the results changed minimally, suggesting that viral rpM did not significantly affect the differential protein expression between SOT recipients and controls (Supplementary Fig. 5a). We further analyzed the relationship between SARS-CoV-2 rpM and protein expression in each of the two study groups (Supplementary Fig. 5b), and found a positive correlation between viral rpM and CXCL8 in the SOT recipients but not in controls (Supplementary Fig. 5c).

Analysis of longitudinal serum cytokine expression dynamics revealed that the IFN-inducible chemokine CXCL11 decreased significantly over time in controls, but not in SOT recipients ($P = 0.0042$, Fig. 4c). After adjusting for viral rpM differences, CXCL11 dynamics remained significantly different between the two groups, along with a more rapid rise over time in CCL3 and CCL4 expression in the SOT recipients compared to the controls (Supplementary Fig. 5d).

### PBMC gene expression differences upon hospitalization, and over time

At the time of hospital admission, differential expression analysis revealed 1047 differentially expressed genes ($P_{adj}$ <0.05) between SOT recipients and controls (Fig. 5a and Supplementary Data 1). Gene set enrichment analysis (GSEA) demonstrated that SOT recipients had increased expression of innate immunity pathways related to type I IFN, TLR signaling, complement activation, IL-1 signaling, and other functions (Fig. 5b). SOT recipients also exhibited lower expression of B-cell receptor signaling and cell cycle-related pathways (Fig. 5c). Adjusting for SARS-CoV-2 rpM in the differential expression analysis demonstrated that the increased expression of complement activation, type I IFN and IL-1 signaling pathways were independent of viral rpM (Supplementary Fig. 6a).

We next evaluated the dynamics of gene expression over the course of hospitalization in SOT recipients and controls. SOT recipients exhibited increased expression over time of genes related to several immune pathways including types I and II IFN signaling, IL-10 and PD-1 signaling, and CD28 co-stimulation (Fig. 5d and Supplementary Data 2). Adjusting for viral rpM did not significantly affect results (Supplementary Fig. 6b). Some signaling pathways (e.g., interferon

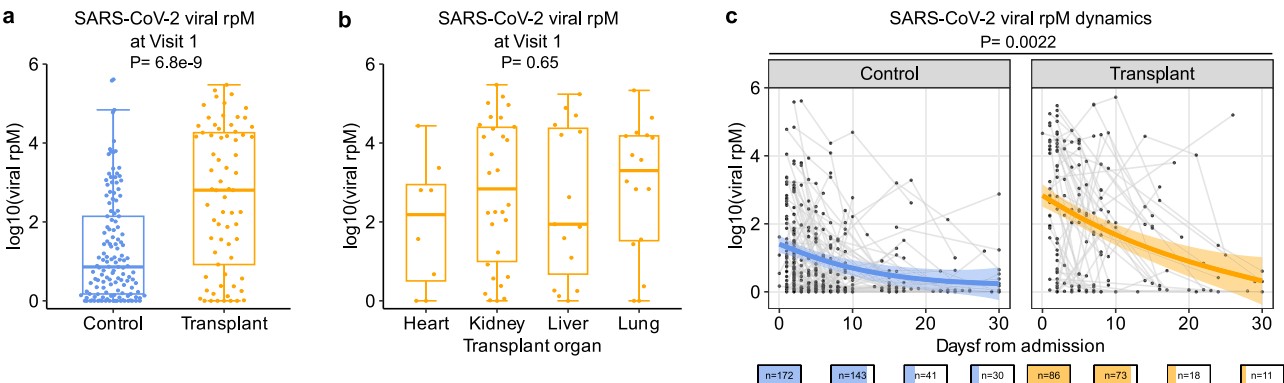

**Fig. 2 | SOT recipients have higher SARS-CoV-2 viral rpM and impaired viral clearance compared to controls. a**, **b** Box plots showing SARS-CoV-2 viral reads per million (rpM) at Visit 1 of **a** transplant (yellow, $n = 86$) and control groups (blue, $n = 172$), and **b** different organ transplant types (heart−$n = 10$, kidney−$n = 41$, liver−$n = 14$, lung−$n = 17$). P values were calculated with **a** a linear model or **b** two-sided likelihood ratio test. Boxes show the median and interquartile range (IQR), whiskers were calculated as the 25th percentile minus 1.5 times the IQR and the 75th percentile plus 1.5 times the IQR. **c** Plot showing the dynamics of viral rpM up to 30 days after hospital admission of the transplant and control groups. The blue and orange lines indicated the generalized additive mixed model fits, and the ribbons indicated the 95% confidence interval of the fits. P value was calculated for the interaction between SOT status and days from admission with a generalized additive mixed model. The number of patients sampled at each time point is depicted graphically below the X axis of (**c**).

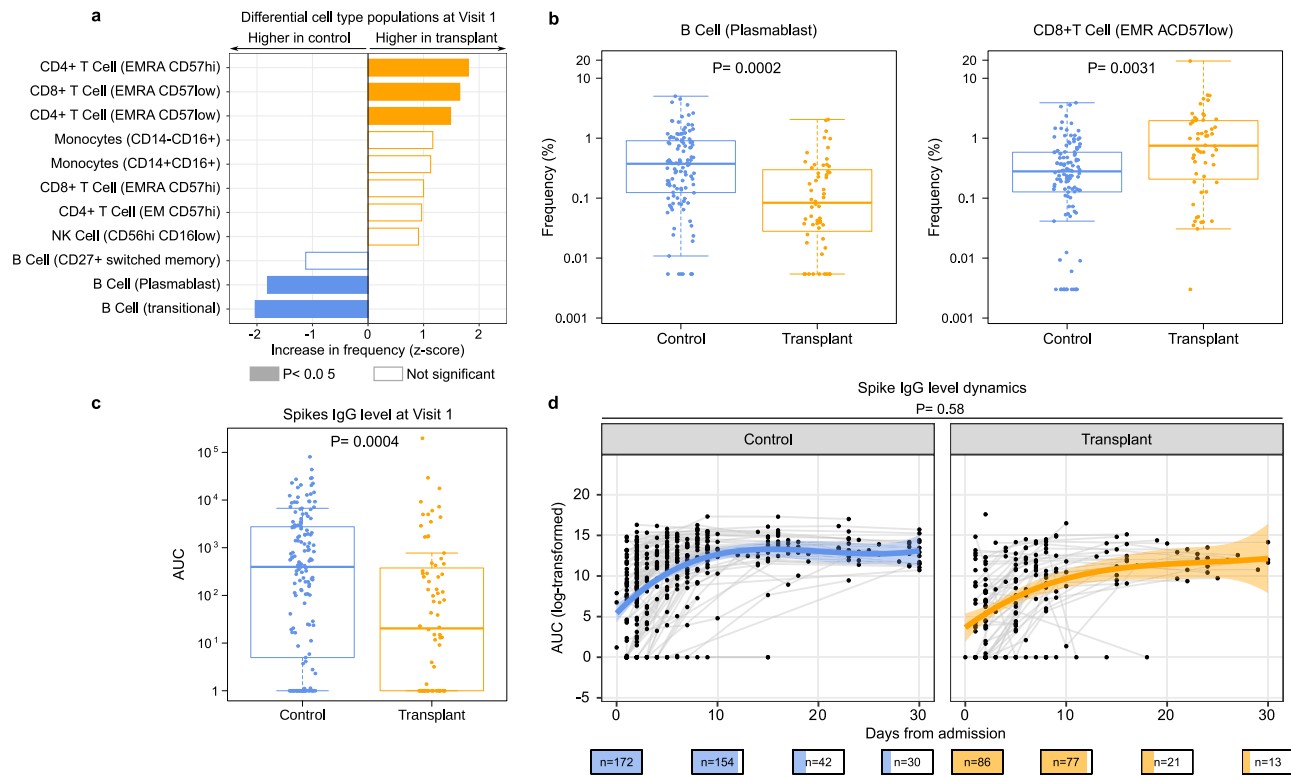

**Fig. 3 | Compared to controls, SOT recipients have lower B-cell plasmablasts and higher EMRA T cells as well as lower SARS-CoV-2 antibody levels at hospitalization. a** Differences in immune cell population frequency measured by CyTOF by SOT recipients (yellow, $n = 54$) and controls (blue, $n = 107$). **b** Box plots highlighting two cell types which differed in frequency between SOT recipients and controls. Boxes show the median and interquartile range (IQR), whiskers were calculated as the 25th percentile minus 1.5 times the IQR and the 75th percentile plus 1.5 times the IQR. *P* values in (**a**, **b**) were calculated with a linear model and Benjamini−Hochberg correction. **c** Box plot of spike IgG levels measured by area under the curve (AUC) by SOT recipients ($n = 86$) and controls ($n = 172$). **d** Longitudinal dynamics of spike IgG levels (log-transformed AUC) in SOT recipients and controls over the course of hospitalization. The blue and orange lines indicate the generalized additive mixed model fits, and the ribbons indicate the 95% confidence interval of the fits. *P* values were calculated with **c** a linear model or **d** a generalized additive mixed model. The number of patients sampled at each time point is depicted graphically below the X axis of (**d**). EMRA effector memory re-expressing CD45RA.

signaling) decreased more strongly over time in non-SOT controls compared to SOT recipients (Fig. 5e). For other pathways (e.g., platelet activation, signaling and aggregation), SOT recipients demonstrated pathway upregulation over time, while in the control group down-regulation was observed (Fig. 5e and Supplementary Data 3).

## Upper respiratory tract gene expression differences between SOT recipients and controls

Recognizing that the respiratory tract is the site of active infection in COVID-19, we performed gene expression analyses from nasal swab specimens. Surprisingly, despite the significant difference in

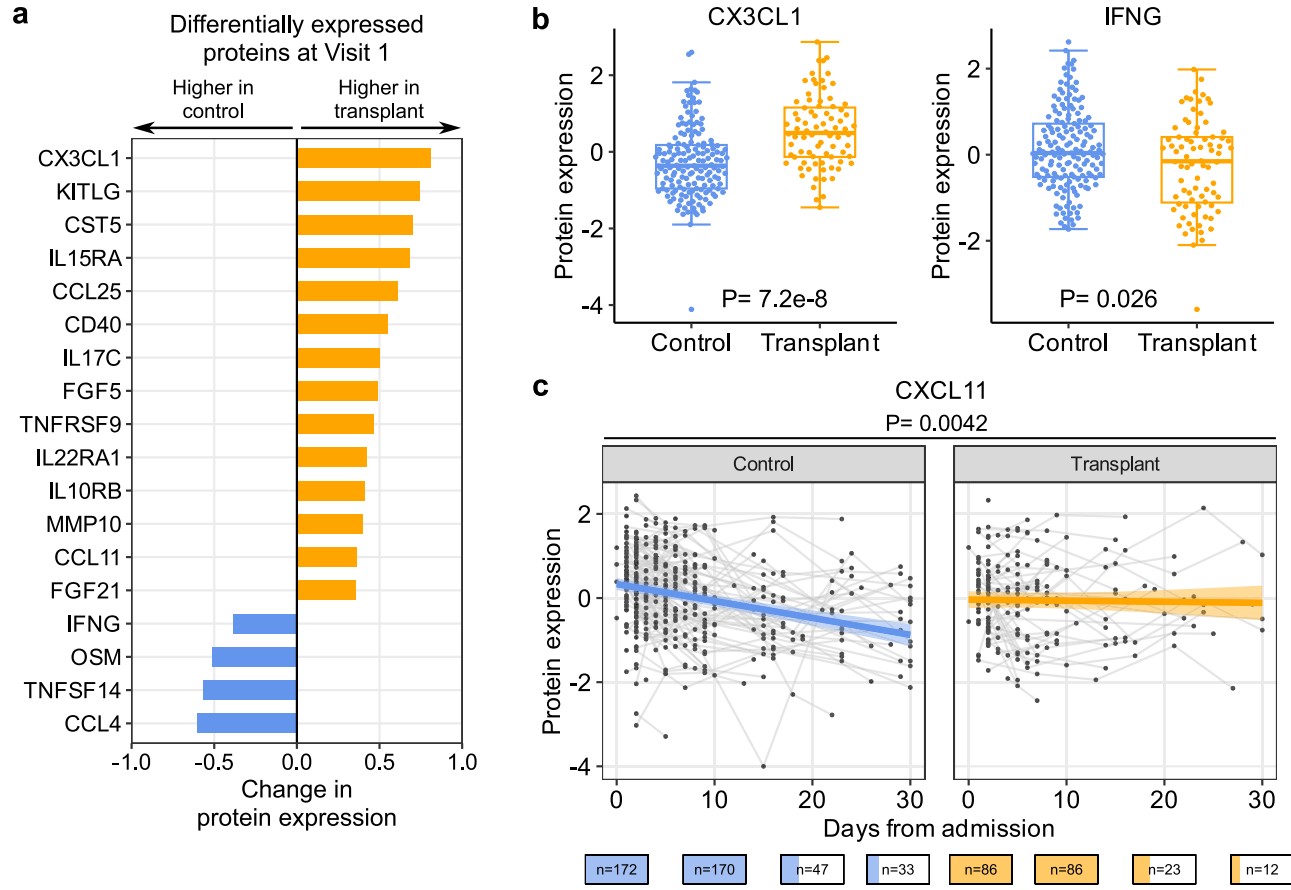

**Fig. 4 | SOT recipients have higher levels of specific serum chemokines and lower levels of IFN-gamma. a** Bar plots showing proteins that are differentially expressed between control (blue, $n = 161$) and transplant patients (yellow, $n = 80$) at Visit 1 (adjusted $P < 0.05$). **b** Box plots showing the levels of CX3CL1 and IFNG at Visit 1. **a**, **b** $P$ values were calculated using a linear model and Benjamini–Hochberg correction. Boxes show the median and interquartile range (IQR), whiskers were calculated as the 25th percentile minus 1.5 times the IQR and the 75th percentile plus 1.5 times the IQR. **c** Scatter plot showing the dynamics of CXCL11 level after hospital admission (without adjusting for SARS-CoV-2 viral rpM). The ribbons indicate the 95% confidence interval of the linear mixed-effects model fits. $P$ value was calculated using a linear mixed-effects model and Benjamini–Hochberg correction. The number of patients sampled at each time point is depicted graphically below the $X$ axis of (**c**).

the nasal SARS-CoV-2 rpM (Fig. 2a), no differentially expressed genes were identified between the two groups at FDR < 0.05 at the time of hospital admission. GSEA nonetheless demonstrated that SOT recipients exhibited increased expression of genes related to IL-10 signaling, neutrophil degranulation, type I IFN signaling, IL-1 and IL-4/IL-13 signaling in the upper respiratory tract at the time of hospital admission (Fig. 6a and Supplementary Data 4), mirroring to some extent our observations in the blood. Most inflammatory pathways differentially upregulated in SOT recipients were unaffected by viral rpM adjustment (Supplementary Fig. 7a).

Longitudinal nasal transcriptional profiling analyses demonstrated increased expression over time of genes related to IFN signaling, TCR signaling, and other immune signaling pathways in SOT recipients (Fig. 6b, c). In contrast, non-SOT controls demonstrated increased expression over time of genes related to neutrophil degranulation and IL-36 signaling (Fig. 6b and Supplementary Data 5). Adjusting for viral load did not meaningfully change results (Supplementary Fig. 7b).

Taken together, these results suggested that SOT recipients, in both the upper respiratory tract and the blood compartments, exhibit augmented innate immune responses at the transcriptional level compared to non-SOT controls, with some compartment-specificity to the relevant immune signaling pathways.

### Differing relationships between interferon signaling and viral abundance in SOT recipients versus controls

In both the blood and the upper respiratory tract, SOT recipients exhibited increased type I IFN gene expression in a viral rpM-independent manner (Supplementary Figs. 6a and 7a). We further explored this by comparing the relationship between IFN-stimulated gene (ISG) expression and viral rpM in SOT recipients versus non-SOT controls (Supplementary Fig. 8). In the blood, ISG expression strongly correlated with viral rpM in non-SOT controls, but this relationship was weaker in the SOT recipients (Supplementary Fig. 8a, c). In contrast, in the upper respiratory tract, ISGs correlated with viral rpM in both groups (Supplementary Fig. 8b, d). Beyond type I and type II IFN signaling, we also found that TLR signaling, neutrophil degranulation, and other immune signaling pathways in the upper airway correlated with SARS-CoV-2 rpM in both SOT recipients and controls (Supplementary Fig. 9).

### Airway microbiome differences between SOT recipients and controls

Next, we used nasal metatranscriptomics to assess whether the composition of the respiratory microbiome differed between SOT recipients and controls upon hospital admission. We found that SOT recipients had greater upper airway microbiome alpha diversity, as measured by the Shannon Diversity Index (SDI),

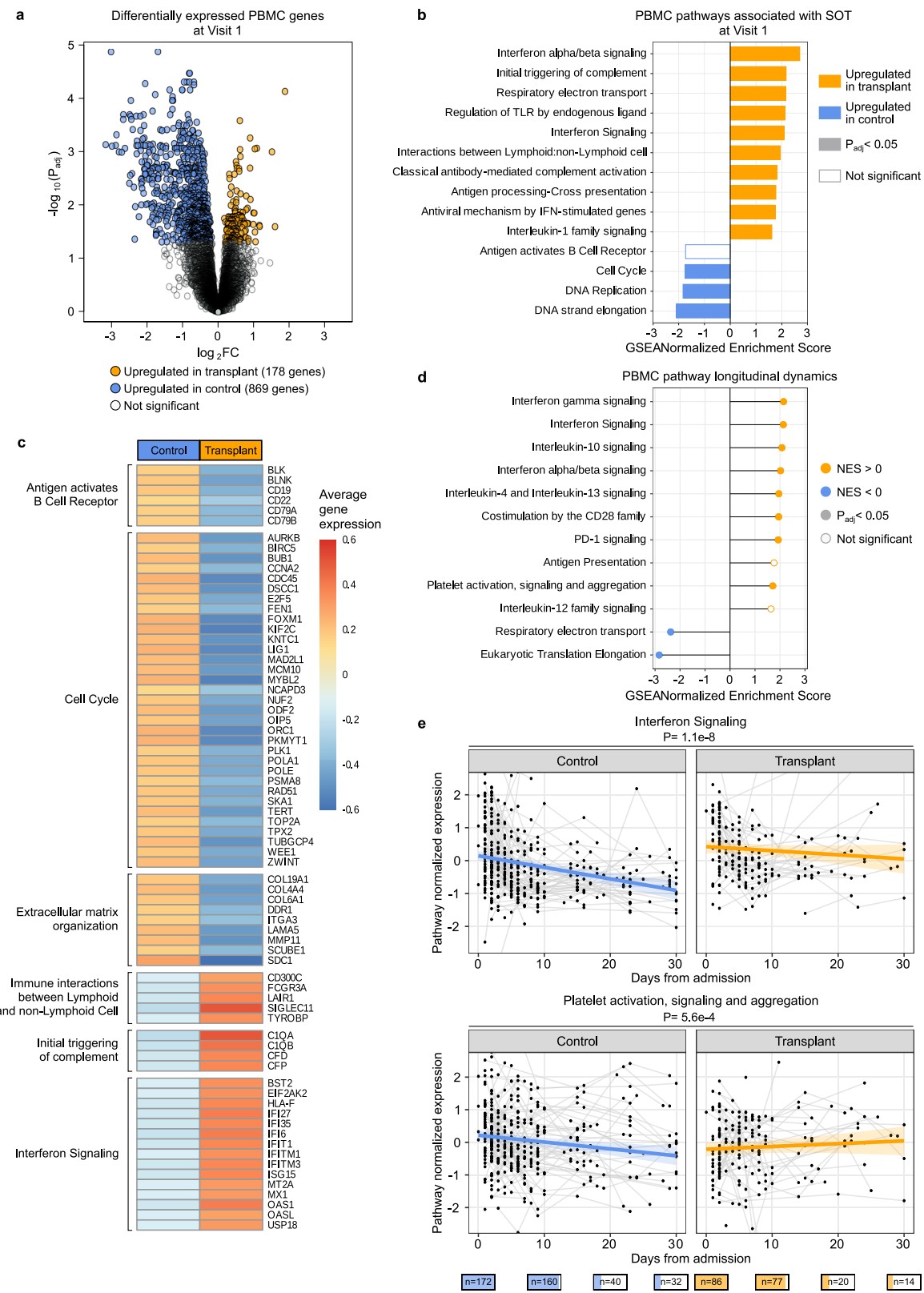

compared to controls at Visit 1 (Fig. 6d, e). No differences in bacterial community composition (beta diversity) between groups, as measured by the Bray–Curtis Dissimilarity Index, were found at Visit 1 ($P = 0.186$, Supplementary Fig. 10). Finally, we asked whether specific taxa differed between groups, and found that only SARS-CoV-2 was significantly more abundant in the SOT recipients (Fig. 6f).

## Immune correlates of COVID-19 severity differ between SOT recipients and controls

We characterized differences in host correlates of COVID-19 severity[15–17] between SOT recipients and non-SOT controls by comparing these groups with respect to cell-type frequencies, gene expression, and protein expression differences between patients with severe COVID-19 (TG 4–5) versus those with mild/moderate COVID-19 (TG 1–3).

**Fig. 5 | PBMC transcriptomics demonstrates that SOT recipients exhibit increased innate immune gene expression upon hospitalization, and over time. a** Volcano plot highlighting genes differentially expressed ($P_{adj}$ <0.05) between SOT recipients (yellow, $n = 66$) and controls (blue, $n = 147$) at the time of hospitalization. **b** gene set enrichment analysis (GSEA) highlighting pathways differentially enriched in SOT recipients versus controls (without adjustment for SARS-CoV-2 viral reads per million (rpM)). A positive normalized enrichment score (NES) value indicates that the pathway was enriched over time in SOT. A negative NES value indicates that the pathway was enriched over time in controls. **c** Average gene expression plot of leading-edge genes from significant GSEA pathways.

**d** Differences in the longitudinal dynamics of signaling pathways. **e** Longitudinal plots highlighting changes in normalized expression of representative immune signaling pathways that significantly differed over time in SOT recipient versus controls. The blue and orange lines indicated the linear mixed-effects model fits, and the ribbons indicate the 95% confidence interval of the fits. *P* values were calculated with **a** a linear model or (**b**, **d**, **e**) a linear mixed-effects model with Benjamini–Hochberg correction. The number of patients sampled at each time point is depicted graphically below the *X* axis of (**d**). PBMC peripheral blood mononuclear cells, SOT solid organ transplant.

In both SOT recipients and controls, severe disease was characterized by reductions in several immune cell populations, including conventional dendritic cells (DCs), intermediate (CD14 + CD16 + ) monocytes, and several CD4 + T-cell subsets, as has been previously observed[17,20] (Fig. 7a). Only SOT recipients, however, had significantly lower CD8+ central memory T cells, CD11C + CXCR5- B cells, and CD56[high] CD16[low] NK cells in severe disease (Fig. 7a). Severe COVID-19 in controls, but not SOT recipients, was associated with a marked increase in several canonical proinflammatory serum cytokines and chemokines (e.g., IL-6, CCL23, and CXCL8) (Fig. 7b). Conversely, serum levels of IFNG and IL12B were significantly lower in severe COVID-19 among SOT recipients, but not among controls.

A similar analysis of PBMC transcriptomics data revealed that both SOT recipients and controls exhibited greater expression of several immune signaling pathways in severe disease, including neutrophil degranulation, innate immune system signaling, IL-1 signaling, and cellular responses to stress (Fig. 7c). The expression of PBMC genes related to PD-1 signaling decreased in severe COVID-19 compared to mild/moderate COVID-19 in both groups as well. SOT recipients, however, demonstrated lower expression of genes related to TCR signaling, CD28 signaling, and phosphorylation of CD3 and TCR zeta chain (Fig. 7c).

Even more notable differences between groups were observed in severity analyses of the upper airway data. For instance, severe disease in controls was characterized by increased expression of genes related to Toll-like receptor (TLR) signaling, whereas in SOT recipients this was not observed (Fig. 7d). Non-SOT controls demonstrated increases in expression of genes related to neutrophil degranulation, IL-10, IL-4/IL-13, and innate immune signaling.

## Discussion

Pharmacologic immunosuppression is necessary to prevent rejection following SOT, but comes at the expense of increased vulnerability to infection. While it is well known that SOT recipients can exhibit clinically atypical responses to respiratory infections including COVID-19[21], the molecular features of these differences have remained unclear. Here, we performed comparative host/microbe systems immunoprofiling of SOT recipients and matched non-SOT controls to address this key knowledge gap. Unexpectedly, we found that COVID-19 in SOT recipients is not characterized by globally suppressed systemic immune signaling, but instead by augmented innate immune responses and more subtle differences across states of COVID-19 severity (Fig. 8).

In the peripheral blood of SOT recipients, augmented innate immune signaling was characterized by higher expression of genes related to type I IFN, IL-1, and complement system pathways. Throughout the course of hospitalization, SOT recipients demonstrated consistent increases in these inflammatory signaling pathways, as well as in PD-1 and CD28 signaling. At the protein level, SOT recipients had higher levels of a few proinflammatory cytokines, such as CX3CL1, a potent chemoattractant of T cells and monocytes, and KITLG, which plays a role in hematopoiesis. In addition, CXCL11 levels remained elevated over time in SOT recipients, but decreased over time in non-SOT controls. Together, these results highlight an

unexpected state of activated innate immune signaling in SOT recipients at the time of hospitalization, complemented by stable to increased activity of PD-1 signaling and other pathways related to T-cell signaling and exhaustion over the course of hospitalization.

We found that this state of innate immune activation was driven in part by higher SARS-CoV-2 viral load in SOT recipients, as adjustment for SARS-CoV-2 rpM impacted the magnitude of expression differences for some proinflammatory signaling pathways and cytokines. In longitudinal analyses, however, even after adjusting for viral rpM differences, SOT recipients demonstrated consistently greater induction of innate immune signaling pathways in the blood, including type I IFN signaling, compared to non-SOT controls. Furthermore, while ISG expression in the blood strongly correlated with viral rpM in non-SOT controls, this relationship was not consistently observed in SOT recipients, suggesting a partial decoupling between IFN signaling and viral RNA burden. This finding suggests that virus-independent factors may be driving the augmented systemic interferon signaling observed in SOT recipients. Perhaps this reflects non-specific compensatory innate immune activation in the setting of impaired adaptive immunity in SOT recipients.

In the upper airway, transcriptional differences between SOT recipients and controls were subtle, although GSEA did reveal important distinctions between groups at the pathway level. Most notably, as in the blood, SOT recipients demonstrated evidence of upregulated innate immune responses in the airways, characterized by increased expression of genes related to type I IFN signaling, IL-1 signaling, and complement activation. In contrast to the blood, expression of ISGs in the upper airway was strongly correlated with SARS-CoV-2 viral rpM in both non-SOT controls and SOT recipients.

In non-SOT control patients, higher expression of proinflammatory cytokines such as IL-6 correlated with COVID-19 severity, consistent with prior studies[15–17]. In SOT recipients, however, we found that the expression of most inflammatory cytokines minimally differed between mild/moderate and severe disease. In addition, while controls exhibited marked severity-associated increases in the expression of canonical proinflammatory genes, this was not observed in SOT recipients. Instead, severe disease in SOT recipients was associated with lower T-cell signaling gene expression in the blood, as well as less robust induction of TLR signaling pathways in the upper airway (Fig. 7). These observations suggest a profound difference in the immune milieus in SOT versus non-SOT patients, depending on severity. The relatively weak association between increased proinflammatory serum cytokines in SOT patients and severe COVID-19 may have important implications, and suggests that the clinical utility of immune modulatory therapies, such as IL-6 inhibitors (e.g., tocilizumab), or JAK inhibitors (e.g., baricitinib) may not be the same in SOT recipients as in the general population.

Despite their increased susceptibility to SARS-CoV-2 infection and comparatively poor outcomes with other respiratory infections[22], SOT recipients have comparable COVID-19 mortality versus the general population, at least in propensity-matched studies[5,8–10]. Our observation that severe disease in SOT recipients is not characterized by a marked increase in mortality-associated inflammatory cytokines such as IL-6, offers a potential explanation.

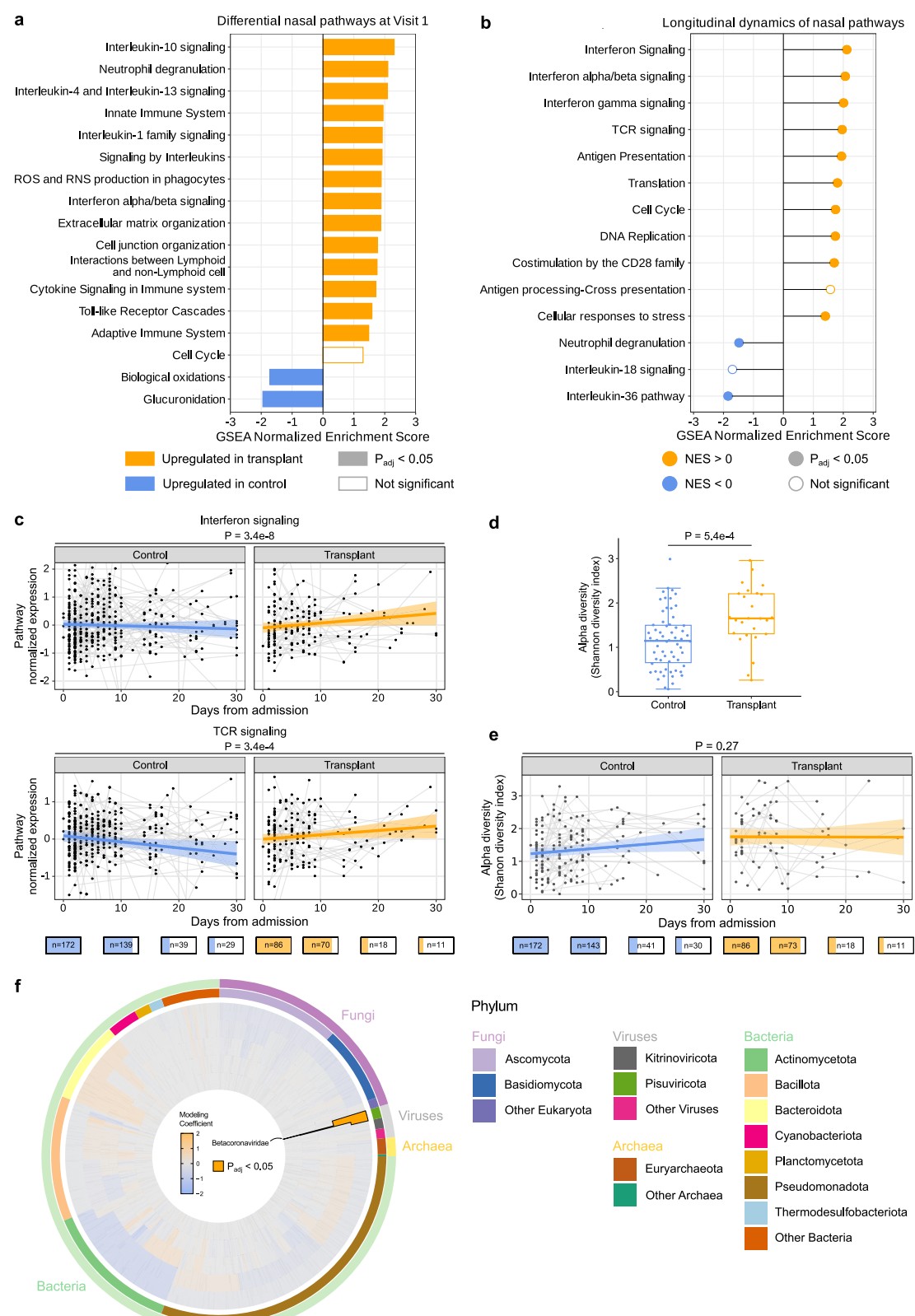

**a** Differential nasal pathways at Visit 1

**b** Longitudinal dynamics of nasal pathways

**c** Interferon signaling — P = 3.4e-8 / TCR signaling — P = 3.4e-4

**d** P = 5.4e-4

**e** P = 0.27

**f** Phylum

Our findings could also simply reflect higher levels of innate immune signaling in SOT recipients versus controls across many states of disease severity, possibly representing a compensatory effect of immunosuppressant medications that predominantly target the adaptive immune system.

These observations lead to a model of immune perturbation in COVID-19 with very different profiles in SOT recipients compared to non-SOT controls. In SOT patients on chronic immune suppression, increased senescent CD4 + T cells and decreased plasmablast and B cells are unable to effectively clear virus, leading to increased and persistent viral replication. Perhaps in a compensatory effort, innate immune responses, such as type I interferon, are upregulated and fail to attenuate appropriately over time. This dysregulated state of impaired B- and T-cell immunity, delayed viral clearance, and

**Fig. 6 | Upper airway host gene expression and the nasal microbiome differ between SOT recipients and controls. a** Gene set enrichment analysis (GSEA) highlighting pathways differentially enriched in solid organ transplant (SOT) recipients (yellow, $n = 63$) versus controls (blue, $n = 125$) in the upper respiratory tract (without adjustment for SARS-CoV-2 viral reads per million (rpM)). **b** Differences in the longitudinal dynamics of signaling pathways. A positive normalized enrichment score (NES) value indicates that the pathway was enriched over time in SOT. A negative NES value indicates that the pathway was enriched over time in controls. **c** Longitudinal plots highlighting changes in normalized expression of representative immune signaling pathways that showed significantly different dynamics in SOT recipients versus controls. The ribbons indicate the 95% confidence interval of the linear mixed-effects model fits. *P* values were calculated with **a** a linear model or **b, c** a linear mixed-effects model with Benjamini–Hochberg correction. **d** Box

plot demonstrating differences in upper airway bacterial microbiome alpha diversity in SOT recipients ($n = 86$) versus controls ($n = 172$). Boxes show the median and interquartile range (IQR), whiskers were calculated as the 25th percentile minus 1.5 times the IQR and the 75th percentile plus 1.5 times the IQR. P values were calculated with the two-sided Wilcoxon rank-sum test. **e** Robust regression with 95% confidence intervals highlighting the longitudinal changes in upper airway alpha diversity following hospitalization. **f** Radial plot highlighting differential abundance from genus (inner most ring) to phylum (outer most ring) and phylogenetic relatedness (inner tree) of taxa differentially enriched in SOT recipients versus controls. *P* values in (**e, f**) were calculated with a linear mixed-effects model and **f** Benjamini–Hochberg correction). The number of patients sampled at each time point is depicted graphically below the *X* axis of (**c, e**).

augmented innate immune signaling has parallels with aging-associated inflammatory changes[23], and may have important implications for the management of immunosuppression in SOT patients with acute infection.

Our study has several strengths. These include a large, comparative immunoprofiling study of vaccine-naïve SOT recipients during their first encounter with a novel viral pathogen without the complication of variable vaccination histories. In addition, our cohort spans multiple medical centers, and assesses both host and microbe using a diverse range of assays are strengths. Our study also has limitations including an insufficient sample size to assess differences based on type of transplanted organ, limited clinical data regarding immunosuppressant dosing, a small sample size of intubated patients with severe COVID-19, a lack of data from the primary site of infection in the lower airway, and a lack of data specific to allograft function. In addition, our assessment of longitudinal trajectories was limited by a reduced number of patients still hospitalized at later timepoints in the study. Therefore, we primarily focused on findings at Visit 1, and our longitudinal findings should be interpreted with this in mind. Finally, further work is needed to determine whether our findings in SOT recipients with COVID-19 also apply to other types of viral, bacterial, and fungal respiratory infections.

Taken together, we find that COVID-19 in SOT recipients is characterized by a biologically distinct immune state with augmented innate signaling but lower proportions of certain adaptive immune cell populations. The distinct immune state of SOT recipients lacks the dynamic induction of genes and cytokines associated with severe COVID-19 in the general population, suggesting a role for prognostic biomarkers and therapeutic approaches in this vulnerable population.

## Methods

### Patient enrollment and sample collection
This study leveraged data from the IMPACC cohort[18,19], which enrolled 1286 participants from 20 hospitals across 15 medical centers in the United States between May 5th, 2020 and March 19th, 2021. Eligible participants were participants hospitalized with SARS-CoV-2 infection confirmed by RT-PCR and symptoms or signs consistent with COVID-19. Solid organ transplant (SOT) patients were identified by review of medication list for immunosuppressive medications. Patients identified as SOT recipients were confirmed by chart review to verify transplant status and organ type. We conducted a case-control study of patients within the IMPACC cohort, matching all 86 solid organ transplant (SOT) recipients in the cohort 2:1 by age, sex and study site with 172 immunocompetent controls. Detailed clinical assessments and sampling of blood and upper respiratory tract were performed within ~72 hours of hospitalization (Visit 1), and on approximately Days 4, 7, 14, 21, and 28 after hospital admission (Supplementary Fig. 11). Biological sample collection and processing followed a standardized protocol[19] across all study sites, and transcriptomic, proteomic, CyTOF, and serologic data were generated at core laboratories.

### Ethics
NIAID staff conferred with the Department of Health and Human Services Office for Human Research Protections (OHRP) regarding potential applicability of the public health surveillance exception [45CFR46.102 (l) (2)] to the IMPACC study protocol. OHRP concurred that the study satisfied criteria for the public health surveillance exception, and the IMPACC study team sent the study protocol, and participant information sheet for review, and assessment to institutional review boards (IRBs) at participating institutions. Twelve institutions elected to conduct the study as public health surveillance, while three sites with prior IRB-approved biobanking protocols elected to integrate and conduct IMPACC under their institutional protocols (University of Texas at Austin, IRB 2020-04-0117; University of California San Francisco, IRB 20-30497; Case Western reserve university, IRB STUDY20200573) with informed consent requirements. Participants enrolled under the public health surveillance exclusion were provided information sheets describing the study, samples to be collected, and plans for data de-identification, and use. Those that requested not to participate after reviewing the information sheet were not enrolled. Participants did not receive compensation for study participation while hospitalized, and subsequently were offered compensation during outpatient follow-up.

### Common statistical analyses framework
Deidentified quality assured raw data was obtained from the IMPACC study and made publicly available[17–19]. All data analyses employed R v4.0.2. For each data type, we investigated the behavior of features both at Visit 1 and longitudinally for scheduled visits (Visits 1–6, up to 30 days post-hospital admission, both inpatient and outpatient samples, and excluding eight additional samples (seven controls, one SOT recipient) collected when a patient was transferred from the ward to intensive care unit. Five COVID-19 severity trajectory groups (TG) were identified by latent class modeling of longitudinal measures of a 7-point clinical severity ordinal scale[17,18]. TG 1 was characterized by relatively mild respiratory disease and a brief hospital stay, while TG 2–4 represented patients with increasing respiratory support requirements and longer hospital stays, and TG 5 represented patients with severe COVID-19 that led to death within 28 days[17,18]. For the severity analysis, we defined mild/moderate participants as those with TG 1–3, and severe participants as those with TG 4–5.

For longitudinal analysis of SARS-CoV-2 nasal viral rpM and serum anti-Spike IgG, we used generalized additive models with mixed effects from the package gamm4 (v0.2.6). Generalized additive modeling was preferred for these features due to their clearly non-linear trajectories. For all other data types, we used linear mixed-effects models from the package lme4 (v1.1.25). *P* values in all analyses were adjusted with Benjamini–Hochberg correction. In addition, to confirm the robustness of key longitudinal analyses for viral load, Olink cytokines and PBMC genes, we performed permutation analysis[24] using 1000 iterations (randomly permuting the patient's transplant/control group assignment 1000 times, and then comparing the observed test statistic to this

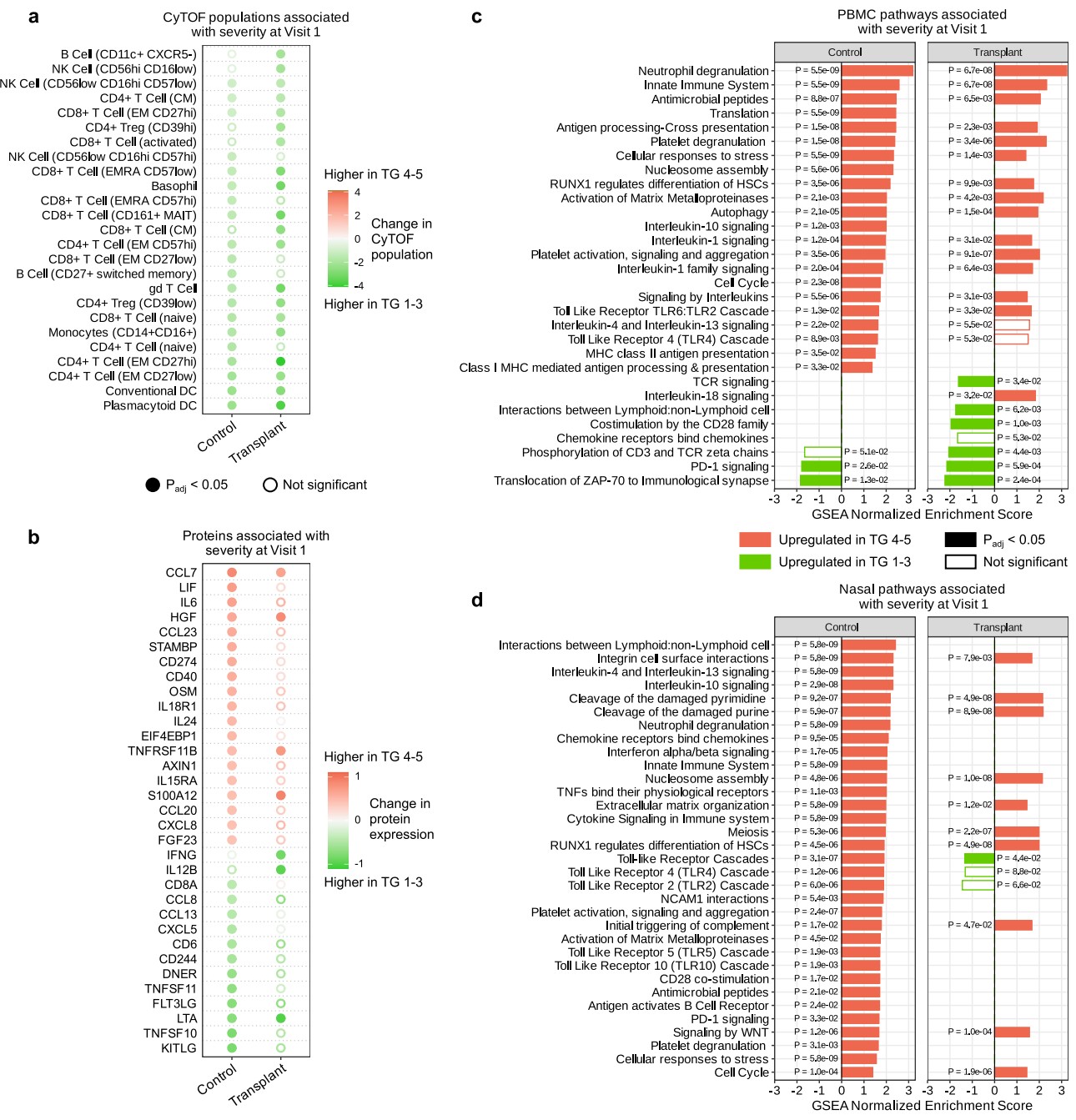

**Fig. 7 | Host immune correlates of COVID-19 severity differ between SOT recipients and controls. a** Dot plot of immune cell populations that are up- or downregulated in severe patients (TG 4–5, red) compared to mild/moderate patients (TG 1–3, green) within each of the control ($n = 107$) and transplant ($n = 54$) groups. **b** Dot plot of proteins that are up- or downregulated in severe compared to mild/moderate patients within each of the control ($n = 161$) and transplant ($n = 80$) groups. **c** Plots highlighting signaling pathways identified by gene set enrichment analysis (GSEA) from peripheral blood mononuclear cell (PBMC) transcriptomics that were differentially upregulated in severe versus mild/moderate COVID-19 in solid organ transplant (SOT) recipients (right, $n = 66$) or controls (left, $n = 147$). **d** Plots highlighting GSEA-identified signaling pathways from nasal transcriptomics that were differentially upregulated in severe versus mild/moderate COVID-19 in SOT recipients (right, $n = 63$) or controls (left, $n = 125$). $P$ values for all analyses were calculated with a linear model and Benjamini–Hochberg correction. CyTOF cytometry by time of flight, PBMC peripheral blood mononuclear cells.

distribution to assess its significance), and calculated the $P$ value as follows:

$$\text{permuted} - P = (\text{number of iterations with test statistic more extreme than the observed test statistic})/1000$$

For each of these three validation analyses, our findings remained statistically significant (permuted-$P < 0.01$).

## Analysis of nasal metatranscriptomics data
Taxonomic alignments from nasal metatranscriptomics data were obtained from raw fastq files using the CZ-ID pipeline[25], which first removes human sequences via subtractive alignment against human genome build 38, followed by quality and complexity filtering. Subsequently, reference-based taxonomic alignment at both the nucleotide and amino acid levels against sequences in the National Center for Biotechnology Information (NCBI) nucleotide (NT) and non-redundant

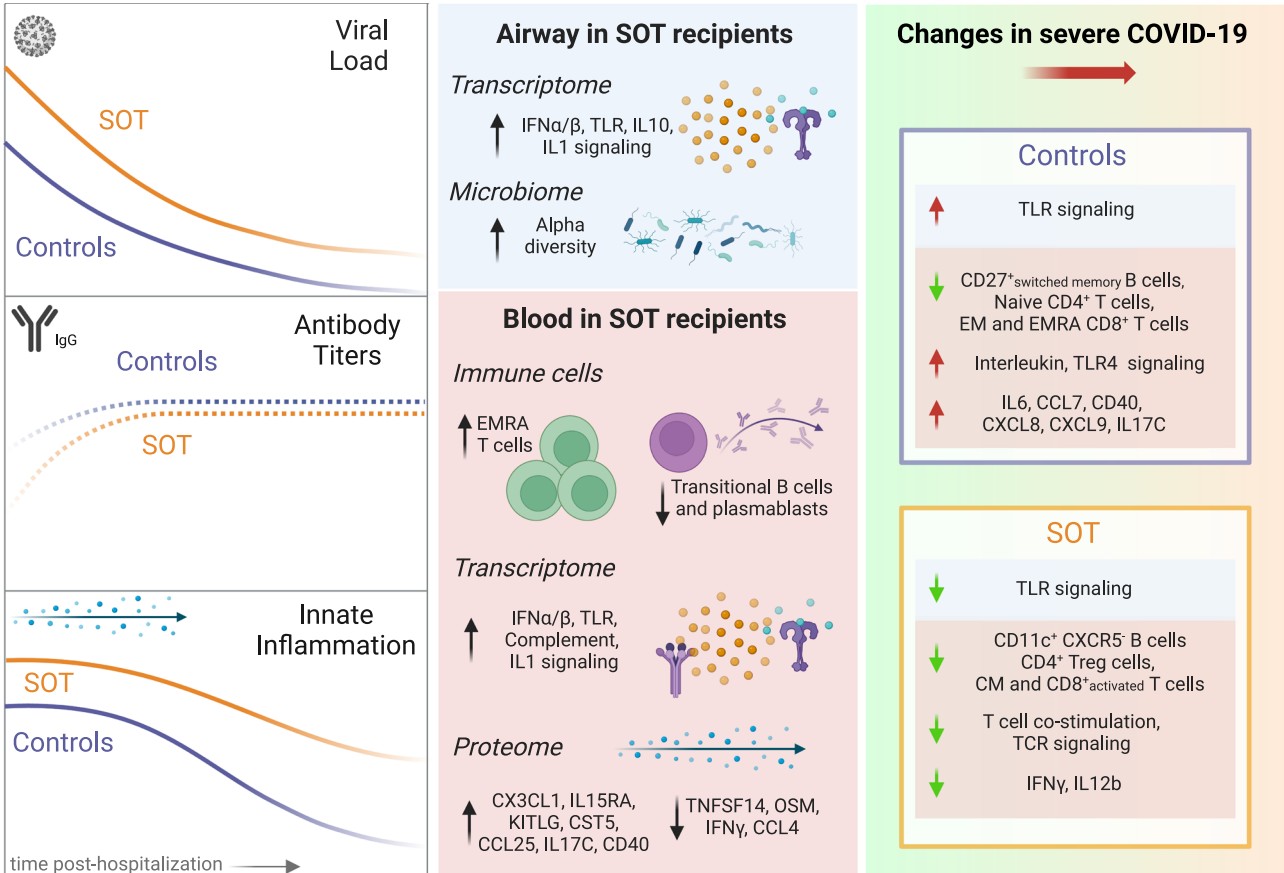

**Fig. 8 | Summary schematic highlighting inflammatory dysregulation in SOT recipients hospitalized for COVID-19 based on host/microbe multiomic profiling.** At the time of hospital admission, solid organ transplant (SOT) recipients had higher SARS-CoV-2 abundance, lower anti-SARS-CoV-2 antibody titers, and augmented innate immune gene and protein expression compared to non-SOT controls. Over time, SOT recipients had impaired viral clearance and exhibited persistently increased expression of innate immune signaling pathways. In the upper airway, SOT recipients exhibited differences in the microbiome and transcriptome. In the blood, SOT recipients demonstrated differences in immune cell populations as well as in the expression of genes and proteins central to innate immune responses. Severe disease in transplant recipients was characterized by a less robust induction of proinflammatory genes and chemokines, as well as by differences in immune cell populations. EMRA effector memory re-expressing CD45RA, EM effector memory, CM central memory. Created in BioRender[33].

(NR) databases, respectively, is carried out, followed by assembly of the reads matching each taxon. Taxa were aggregated to the genus and higher phylogenetic levels from NCBI for analyses[26]. For all analyses using SARS-CoV-2 viral rpM, log transformation of total reads per million (rpM) aligned to the Beta-coronavirus genus was used. Alpha diversity (Shannon Diversity Index) was calculated using the vegan package v.2.6 in R. Differential abundance analyses for Visit 1 samples were performed using linear mixed effect modeling (using the R package nlme v3.1-162) to evaluate SOT effect on individual taxon levels at the genus, family, class, order, phylum, and superphylum levels (rpMs from lower taxon levels were summed to create higher phylogenetic level rpM), using the following R formula:

Taxon_abundance ~ transplant_status + (1|enrollment_site)

In addition, to confirm the finding from linear mixed effect modeling that Betacoronavirus was the only taxa with significant relative abundance changes in SOT recipients, we analyzed Visit 1 samples with "Analysis of Compositions of Microbiomes with Bias Corrections" (ANCOM-BC)[27] which also identified Betacoronavirus as the only significant differentially abundant taxon. Principle coordinate analysis (PCoA) of the Bray−Curtis dissimilarity index was performed on Visit 1 nasal metatranscriptomics samples, with significance calculated with Adonis using the R package vegan (v2.6). Alpha diversity was calculated based on the Shannon Diversity Index:

$$H' = -\sum_{i=1}^{s} p_i \ln(p_i) \qquad (1)$$

Where s is the number of species and $p_i$ is the proportional abundance of species $i$. Beta diversity was calculated based on the Bray−Curtis Dissimilarity Index:

$$BC_{jk} = 1 - \frac{\sum \left| x_{ij} - x_{ik} \right|}{\sum \left( x_{ij} + x_{ik} \right)} \qquad (2)$$

Where $x_{ij}$ and $x_{ik}$ represent the quantity of species ($i$) and two sites ($j$ and $k$).

**Analysis of SARS-CoV-2 viral abundance**
SARS-CoV-2 viral abundance was calculated as log10 (rpM+1), where rpM is the reads per million of SARS-CoV-2 as measured by nasal metatranscriptomics. The viral rpM in each organ transplant type was compared using a likelihood ratio test on the null and

alternative models in R:

$$\text{Null} : \text{viral\_rpm} \sim 1$$

$$\text{Alternative} : \text{viral\_rpm} \sim \text{organ\_type}$$

Where viral_rpm was the log10-transformed viral rpM, and organ_type was the organ transplant type. Longitudinal analysis of SARS-CoV-2 viral rpM was performed using the gamm4 function from the gamm4 package (v0.2.6), using the following R formula:

$$\text{viral\_rpm} \sim s(\text{event\_date}, \text{bs} = ''\text{cr}'')$$
$$+ s(\text{event\_date}, \text{bs} = ''\text{cr}'', \text{by} = \text{transplant}) + \text{transplant}$$

with random effects (1|pid), ie, participant random intercept. In the formula, viral_rpm was the log10-transformed viral rpM as described above, s(event_date, bs = "cr") was the cubic spline fit on the number of days post hospitalization, s(event_date, bs = "cr", by=transplant) was the cubic spline fit on the interaction between the number of days post hospitalization and transplant status, and transplant is the transplant status. *P* value was calculated using the Chi-squared test on the gam component's reference degrees of freedom and F-statistics.

## Analysis of SARS-CoV-2 antibody titers

Antibody levels against the recombinant SARS-CoV-2 spike protein receptor-binding domain (RBD) were measured by enzyme-linked immunosorbent assay (ELISA)[17]. Briefly, following heat inactivation at 56 °C for 1 hour, serum samples were added to plates coated with RBD. Optical density (OD) was measured in a Synergy 4 (BioTek) plate reader at 490 nm. The area under the curve (AUC) was calculated, considering 0.15 OD as the cutoff. For the Visit 1 analysis, we log2-transformed the area under the curve (AUC) values and modeled them with linear regression. For the longitudinal analysis, we also log2-transformed the AUC values, and used the linear mixed-effects models to fit the null and alternative models in R:

$$\text{Null} : z \sim \text{event\_date} + \text{transplant} + (1|\text{pid})$$

$$\text{Alternative} : z \sim \text{event\_date} + \text{transplant} + \text{event\_date} : \text{transplant}$$
$$+ (1|\text{pid})$$

Where z is the log2-transformed AUC, event_date is the number of days post-admission, transplant is the transplant status, event_date:transplant is the interaction term between event_date and transplant status, and (1|pid) is the participant random intercept. The *P* values were calculated using likelihood ratio test, and adjusted with Benjamini–Hochberg correction. For visualization of longitudinal antibody levels, data were fit to a third-order polynomial.

## Analysis of PBMC and nasal RNA-seq data

Nasal turbinate swabs collected into DNA/RNA shield reagent (Zymo Research) underwent RNA extraction using the Quick DNA/RNA Mag-Bead kit (Zymo Research)[17]. Ribosomal depletion, cDNA synthesis, and library preparation were then carried out using the Total Stranded RNA Prep with Ribo-Zero Plus kit (Illumina), following the manufacturer's instructions[17,18].

In total, $2.5 \times 10^5$ PBMCs homogenized in 200 mL of Buffer RLT (Qiagen) underwent RNA extraction using the Quick RNA MagBead Kit (Zymo Research). Library preparation was then carried out using the SMART-Seq v4 Ultra Low Input RNA Kit (Takara Bio). Barcoded and normalized libraries were pooled prior to loading. Paired-end Illumina sequencing was carried out on a NovaSeq 6000 instrument.

We retained protein-coding genes that had a minimum of 10 counts in at least 70% of the samples. We calculated normalization factors to scale library sizes using the calcNormFactors function from the edgeR package[28] v3.40.2, then normalized the gene counts using the voom function (normalize.method = "quantile") from the limma package[29,30] v3.46.0, fitted a linear model for the gene expression with lmFit function (default settings), calculated the empirical Bayes statistics with eBayes function (default settings), and calculated the *P* values for differential expression controlling for FDR. We controlled for log-transformed viral rpM in certain analyses when indicated.

For longitudinal analyses, we accounted for repeat measures from the same individual using duplicateCorrelation from the limma package, and modeled the interaction between days post-admission and transplant status using the R formula:

$$z \sim \text{event\_date} + \text{transplant} + \text{event\_date} : \text{transplant}$$

Where z is the log-transformed normalized expression count, event_date is the number of days post-admission, transplant is the transplant status, and event_date:transplant is the interaction term between event_date and transplant status.

Fold-change values from all genes (regardless of their adjusted *P* values) in the Visit 1 differential expression analyses, representing the fold-change of transplant patients over control patients, and longitudinal analyses, representing the interaction term of days post-admission and transplant status, were used as the input for Gene Set Enrichment Analysis (GSEA). We used the gsePathway function from the ReactomePA v1.42.0 package to search for enriched pathways in the Reactome database, with minimum and maximum geneSet sizes of 3 and 1000, respectively.

For analysis of the relationship between interferon signaling and viral rpM at Visit 1, we first subset the total PBMC and nasal RNA-Seq data to genes within the Reactome *Interferon Signaling* pathway (R-HSA-913531, *n* = 308). We then split the data by transplant status and modeled the relationship between interferon signaling gene expression and log2-transformed viral rpM for controls and transplant recipients separately, using the approach described above. Additionally, we repeated this analysis for the total nasal RNA-Seq dataset, and the results were used as input for GSEA as described above.

## Analysis of CyTOF data

PBMCs were phenotyped on the Fluidigm Helios mass cytometer using a panel of 46 surface and intracellular markers, and the cell types were annotated using an automated annotation pipeline[17]. Briefly, this involved clustering cells from a single sample into 1000 K-means clusters. Using Clustergrammer2[31], a subset of samples was then manually annotated to create a training dataset. Then, the cosine similarity of every cluster to all possible cell types within the training datasets was calculated, and that cluster was assigned to either its highest similarity score cell type or the greatest consensus cell type across the training datasets[17]. The cluster cell-type annotation was then assigned back to the single cells within that cluster, and the number of cells was calculated for a cell type within a given single sample[17].

Prior to analysis, we removed cells identified as red blood cells, multiplets, debris, and those that were not identifiable with high confidence. These counts were converted to proportions per sample, by dividing each cell-type count by the total cell count. The minimum proportion per cell type across all samples was added to each sample prior to log2-transformation, to avoid taking the logarithm of zeros.

For the Visit 1 analysis, the log2-transformed cell-type proportions were modeled with linear regression. For the longitudinal analysis, the log2-transformed cell-type proportions were modeled with linear

mixed-effects models to fit the null and alternative models in R:

$$\text{Null} : z \sim \text{event\_date} + \text{transplant} + (1|\text{pid})$$

$$\text{Alternative} : z \sim \text{event\_date} + \text{transplant} + \text{event\_date} : \text{transplant} + (1|\text{pid})$$

Where z is the log2-transformed cell-type proportion, event_date is the number of days post-admission, transplant is the transplant status, event_date:transplant is the interaction term between event_date and transplant status, and (1|pid) is the participant random intercept. The P values were calculated using likelihood ratio test, and adjusted with Benjamini–Hochberg correction.

### Analysis of serum inflammatory protein (Olink) data

All samples were processed with the Olink 92-protein multiplex inflammatory panel (Olink Proteomics), according to the manufacturer's instructions[17]. Target protein quantification was performed by real-time microfluidic qPCR via the Normalized Protein Expression (NPX) manager software. Data were normalized using internal controls in every sample, inter-plate control and negative controls, and correction factor and expressed as log2 scale proportional to the protein concentration. For additional quality control, we set any NPX measurements below the assay's limit of detection (LOD) to zero. Next, we excluded proteins that were detected in fewer than 20% of samples, resulting in 84 proteins for analysis.

For the Visit 1 analysis, we standardized the NPX values and modeled them with linear regression in R, with and without adjusting for SARS-CoV-2 viral rpM:

$$z \sim \text{transplant}$$

$$z \sim \text{transplant} + \log 10(\text{viral\_rpM} + 1)$$

Where z is the standardized protein level, transplant is the SOT status, and viral_rpM is the SARS-CoV-2 viral rpM as measured by nasal swab metatranscriptomics. The P values were calculated for the transplant coefficient, and adjusted with Benjamini–Hochberg correction.

For the analysis of protein levels and SARS-CoV-2 rpM, we fit the following linear model for the SOT and the control groups separately in R:

$$z \sim \log 10(\text{viral\_rpM} + 1)$$

For the analysis of protein levels and COVID-19 severity, we fit the following linear model:

$$z \sim \text{transplant} + \text{transplant} : TG$$

Where transplant:TG is the interaction term between SOT status and disease severity. This formulation allows us to find two separate coefficients (i.e., two separate log fold-change values) for the effects of severity, one for the SOT group and one for the control group. The P values were calculated for each of these two coefficients, and adjusted with Benjamini–Hochberg correction.

For the longitudinal analysis, we also standardized the NPX values, and used the linear mixed-effects models to fit the null and alternative models in R:

$$\text{Null} : z \sim \text{event\_date} + \text{transplant} + (1|\text{pid})$$

$$\text{Alternative} : z \sim \text{event\_date} + \text{transplant} + \text{event\_date} : \text{transplant} + (1|\text{pid})$$

Where z is the standardized longitudinal protein level, event_date is the number of days post-hospital admission, transplant is the transplant status, event_date:transplant is the interaction term between event_date and transplant status, and (1|pid) is the participant random intercept. The P values were calculated using a likelihood ratio test, and adjusted with Benjamini–Hochberg correction.

### Analysis of immunosuppressive medications

Mycophenolate and tacrolimus were the most common immunosuppressive medications at admission in this cohort of SOT recipients, being received by 55.8% and 73.3%, respectively. We modeled the relationship between severe disease, defined as trajectory group 4 or 5, and whether SOT recipients were receiving mycophenolate or tacrolimus, independently, with logistic regression. We modeled the relationship between visit-1 nasal SARS-CoV-2 rpM and serum anti-RBD IgG AUC, both as described above, with linear regression.

### Reporting summary

Further information on research design is available in the Nature Portfolio Reporting Summary linked to this article.

## Data availability

The clinical metadata, antibody titer, RT-qPCR, mass cytometry and Olink proteomics data have been deposited in ImmPort under accession number SDY1760. The RNA-sequencing data have been deposited in dbGAP under accession number phs002686 (https://www.ncbi.nlm.nih.gov/projects/gap/cgi-bin/study.cgi?study_id=phs002686.v2.p2). The analyzed data used to generate the figures are provided in the Source Data file. Source data are provided with this paper.

## Code availability

All analysis code has been deposited at https://bitbucket.org/kleinstein/impacc-public-code/src/master/solid_organ_transplant_manuscript/.

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

## Acknowledgements

Clinical and Data Coordinating Center: Sanya Thomas, Mitchell Cooney, Shun Rao, Sofia Vignolo, Elena Morrocchi. David Geffen School of Medicine at the University of California- Los Angeles: Arash Naeim, Marianne Bernardo, Sarahmay Sanchez, Shannon Intluxay, Clara Magyar, Jenny Brook, Estefania Ramires-Sanchez, Megan Llamas, Claudia Perdomo, Clara E. Magyar, Jennifer A. Fulcher, and the UCLA Center for Pathology Research Services and the Pathology Research Portal. Yale School of Medicine: M. Catherine Muenker, Dimitri Duvilaire, Maxine Kuang, William Ruff, Khadir Raddassi, Denise Shepherd, Haowei Wang, Omkar Chaudhary, Syim Salahuddin, John Fournier, Michael Rainone, Maxine Kuang. The study was funded by the United States National Institutes of Health through the following grants: 5R01AI135803-03, 5U19AI118608-04, 5U19 AI128910-04, 5U19 AI089992, 4U19AI090023-11, 4U19AI118610-06, R01AI145835-01A1S1, 5U19AI062629-17, 5U19AI057229-17, 5U19AI125357-05, 5U19AI128913-03, 3U19AI077439-13, 5U54AI142766-03, 5R01AI104870-07, 3U19AI089992-09, 3U19AI128913-03, NIH-NIAID 3U19AI1289130, U19AI128913-04S1, 5R01HL155418 and the Chan Zuckerberg Biohub.

## Author contributions

H.P., J.S., H.V.P., A.T., E.F.R., and C.R.L. conceived the idea for the project. M.C.A., S.E.B., W.E., N.D.J., and S.K-S. generated the data. The I.N. contributed to cohort design, participant enrollment, sample collection, data generation and/or data quality assurance. H.P., H.V.P., A.T., A.H., and C.M. analyzed the data. H.P., J.S., H.V.P., A.T., C.M., N.R., N.I.A.H., M.A.A., S.B., M.F., W.M., R.S., M.C.A., P.M.B., S.E.B., W.E., A.H., N.D.J., S.K., M.J., S.H.K., F.K., H.T.M., A.O., J.D-A., A.S., L.B., O.L., E.F.R., and C.R.L. provided input on analyses and findings. H.P., J.S., H.V.P., C.M., E.F.R., and C.R.L. wrote the manuscript. H.P., J.S., H.V.P., A.T., C.M., N.R., N.I.A.H., M.A.A., S.B., M.F., W.M., R.S., M.C.A., P.M.B., S.E.B., W.E., A.H., N.D.J., S.K., M.J., S.H.K., F.K., H.T.M., A.O., J.D-A., A.S., L.B., O.L., E.F.R., and C.R.L. reviewed and edited the manuscript.

## Competing interests

F.K. has the following financial interests: The Icahn School of Medicine at Mount Sinai has filed patent applications relating to SARS-CoV-2 serological assays, NDV-based SARS-CoV-2 vaccines, influenza virus vaccines, and influenza virus therapeutics which list Florian Krammer as co-inventor (Patent title and number: Influenza Virus Vaccines and Uses Thereof (Chimeric HA 2) 9,371,366; Influenza Virus Vaccines and Uses Thereof (Chimeric HA 1) 10,131,695; Influenza Virus Vaccines and Uses Thereof (Chimeric HA 2) 2934581; Influenza Virus Vaccines and Uses Thereof (Chimeric HA 2) 9,968,670; Influenza Virus Vaccines and Uses Thereof (Chimeric HA 2) 10,137,189, Influenza Virus Vaccines and Uses Thereof (Chimeric HA 2) 10,583,188; Influenza Virus Vaccines and Uses Thereof (Chimeric HA 1) EP2758075; Influenza Virus Vaccination Regimens (Neuraminidase) 10,736,956; Anti-Influenza B Virus Neuraminidase Antibodies and Uses Thereof 11254733; Influenza Virus Hemagluttinin Proteins and Uses Thereof (Mosaic) 7237344). Mount Sinai has spun out a company, Kantaro, to market serological tests for SARS-CoV-2 and another company, Castlevax, to develop SARS-CoV-2 vaccines. F.K. is a co-founder and scientific advisory board member of Castlevax. F.K. has consulted for Merck, Curevac, Seqirus, and Pfizer and is currently consulting for 3rd Rock Ventures, GSK, Gritstone, and Avimex. The Krammer laboratory is also collaborating with Dynavax on influenza vaccine development. R.R.M. has a Leadership Councilor role 2018-2021 for the Society of Leukocyte Biology. O.L. has received support as a speaker for presentation regarding the Coronavirus pandemic from Midsized Bank Coalition of Americ (MBCA) and Moody's Analytics. N.G.R. has research grants from Pfizer, Merck, Sanofi, Quidel, Immorna, Vaccine Company,

and Lilly, serves on safety committees for ICON and EMMES and the advisory boards of Moderna, Seqirus, Pfizer, and Sanofi, and is a paid safety consultant for ICON, CyanVac and EMMES. The remaining authors declare no competing interests.

## Additional information

[1]David Geffen School of Medicine, University of California Los Angeles, Los Angeles, CA, USA. [2]University of California San Francisco, San Francisco, CA, USA. [3]The University of Texas at Austin, Austin, TX, USA. [4]Emory School of Medicine, Atlanta, GA, USA. [5]Oklahoma University Health Sciences Center, Oklahoma City, OK, USA. [6]University of Florida, Gainesville, FL, USA. [7]Oregon Health Sciences University, Portland, OR, USA. [8]Benaroya Research Institute, University of Washington, Seattle, WA, USA. [9]National Institute of Allergy and Infectious Diseases/National Institutes of Health, Bethesda, MD, USA. [10]Precision Vaccines Program, Boston Children's Hospital, Boston, MA, USA. [11]Icahn School of Medicine at Mount Sinai, New York, NY, USA. [12]Yale School of Medicine, New Haven, CT, USA. [13]Stanford University School of Medicine, Palo Alto, CA, USA. [14]Broad Institute of MIT & Harvard, Boston, MA, USA. [15]Harvard Medical School, Boston, MA, USA. [16]Brigham & Women's Hospital, Boston, MA, USA. [17]Chan Zuckerberg Biohub San Francisco, San Francisco, CA, USA. [30]These authors contributed equally: Harry Pickering, Joanna Schaenman, Hoang Van Phan, Cole Maguire. [36]These authors jointly supervised this work: Elaine F. Reed, Charles R. Langelier. ✉e-mail: chaz.langelier@ucsf.edu

## IMPACC Network

Elaine F. Reed[1], Joanna Schaenman[1,30], Ramin Salehi-rad[1], Adreanne M. Rivera[1], Harry Pickering[1], Subha Sen[1], David Elashoff[1], Dawn C. Ward[1], Jenny Brook[1], Estefania Ramires-Sanchez[1], Megan Llamas[1], Claudia Perdomo[1], Clara E. Magyar[1], Jennifer Fulcher[1], David J. Erle[2], Carolyn S. Calfee[2], Carolyn M. Hendrickson[2], Kirsten N. Kangelaris[2], Viet Nguyen[2], Deanna Lee[2], Suzanna Chak[2], Rajani Ghale[2], Ana Gonzalez[2], Alejandra Jauregui[2], Carolyn Leroux[2], Luz Torres Altamirano[2], Ahmad Sadeed Rashid[2], Andrew Willmore[2], Prescott G. Woodruff[2], Matthew F. Krummel[2], Sidney Carrillo[2], Alyssa Ward[2], Charles R. Langelier[2,17,31]✉, Ravi Patel[2], Michael Wilson[2], Ravi Dandekar[2], Bonny Alvarenga[2], Jayant Rajan[2], Walter Eckalbar[2], Andrew W. Schroeder[2], Gabriela K. Fragiadakis[2], Alexandra Tsitsiklis[2], Eran Mick[2], Yanedth Sanchez Guerrero[2], Christina Love[2], Lenka Maliskova[2], Michael Adkisson[2], Aleksandra Leligdowicz[2], Alexander Beagle[2], Arjun Rao[2], Austin Sigman[2], Bushra Samad[2], Cindy Curiel[2], Cole Shaw[2], Gayelan Tietje-Ulrich[2], Jeff Milush[2], Jonathan Singer[2], Joshua J. Vasquez[2], Kevin Tang[2], Legna Betancourt[2], Lekshmi Santhosh[2], Logan Pierce[2], Maria Tecero Paz[2], Michael M. Matthay[2], Neeta Thakur[2], Nicklaus Rodriguez[2], Nicole Sutter[2], Norman Jones[2], Pratik Sinha[2], Priya Prasad[2], Raphael Lota[2], Sadeed Rashid[2], Saurabh Asthana[2], Sharvari Bhide[2], Tasha Lea[2], Yumiko Abe-Jones[2], Lauren I. R. Ehrlich[3], Esther Melamed[3], Cole Maguire[3], Dennis Wylie[3], Justin F. Rousseau[3], Kerin C. Hurley[3], Janelle N. Geltman[3], Nadia Siles[3], Jacob E. Rogers[3], Pablo Guaman Tipan[3], Nadine Rouphael[4], Steven E. Bosinger[4], Arun K. Boddapati[4], Greg K. Tharp[4], Kathryn L. Pellegrini[4], Brandi Johnson[4], Bernadine Panganiban[4], Christopher Huerta[4], Evan J. Anderson[4], Hady Samaha[4], Jonathan E. Sevransky[4], Laurel Bristow[4], Elizabeth Beagle[4], David Cowan[4], Sydney Hamilton[4], Thomas Hodder[4], Amer Bechnak[4], Andrew Cheng[4], Aneesh Mehta[4], Caroline R. Ciric[4], Christine Spainhour[4], Erin Carter[4], Erin M. Scherer[4], Jacob Usher[4], Kieffer Hellmeister[4], Laila Hussaini[4], Lauren Hewitt[4], Nina Mcnair[4], Susan Pereira Ribeiro[4], Sonia Wimalasena[4], Jordan P. Metcalf[5], Nelson I. Agudelo Higuita[5], Lauren A. Sinko[5], J. Leland Booth[5], Douglas A. Drevets[5], Brent R. Brown[5], Mark A. Atkinson[6], Scott C. Brakenridge[6], Ricardo F. Ungaro[6], Brittany Roth Manning[6], Lyle Moldawer[6], William B. Messer[7], Catherine L. Hough[7], Sarah A. R. Siegel[7], Peter E. Sullivan[7], Zhengchun Lu[7], Amanda E. Brunton[7], Matthew Strand[7], Zoe L. Lyski[7], Felicity J. Coulter[7], Courtney Micheletti[7], Matthew C. Altman[8], Naresh Doni Jayavelu[8], Scott Presnell[8], Bernard Kohr[8], Tomasz Jancsyk[8], Azlann Arnett[8], Patrice M. Becker[9], Alison D. Augustine[9], Steven M. Holland[9], Lindsey B. Rosen[9], Serena Lee[9], Tatyana Vaysman[9], Al Ozonoff[10], Joann Diray-Arce[10,15], Jing Chen[10], Alvin T. Kho[10], Carly E. Milliren[10], Annmarie Hoch[10], Ana C. Chang[10],

Kerry McEnaney[10], Caitlin Syphurs[10], Brenda Barton[10], Claudia Lentucci[10], Maimouna D. Murphy[10], Mehmet Saluvan[10], Tanzia Shaheen[10], Shanshan Liu[10], Marisa Albert[10], Arash Nemati Hayati[10], Robert Bryant[10], James Abraham[10], Mitchell Cooney[10], Meagan Karoly[10], Ofer Levy [10,14,15], Hanno Steen[10], Patrick van Zalm[10], Benoit Fatou[10], Kinga K. Smolen[10], Arthur Viode[10], Simon van Haren[10], Meenakshi Jha[10], David Stevenson[10], Sanya Thomas[10], Boryana Petrova[10], Naama Kanarek[10], Ana Fernandez-Sesma[11], Viviana Simon[11], Florian Krammer [11], Harm Van Bakel[11], Seunghee Kim-Schulze[11], Ana Silvia Gonzalez-Reiche[11], Jingjing Qi[11], Brian Lee[11], Juan Manuel Carreño[11], Gagandeep Singh[11], Ariel Raskin[11], Johnstone Tcheou[11], Zain Khalil[11], Adriana van de Guchte[11], Keith Farrugia[11], Zenab Khan[11], Geoffrey Kelly[11], Komal Srivastava[11], Lily Q. Eaker[11], Maria C. Bermúdez-González[11], Lubbertus C. F. Mulder[11], Katherine F. Beach[11], Miti Saksena[11], Deena Altman[11], Erna Kojic[11], Levy A. Sominsky[11], Arman Azad[11], Dominika Bielak[11], Hisaaki Kawabata[11], Temima Yellin[11], Miriam Fried[11], Leeba Sullivan[11], Sara Morris[11], Giulio Kleiner[11], Daniel Stadlbauer[11], Jayeeta Dutta[11], Hui Xie[11], Manishkumar Patel[11], Kai Nie[11], Brian Monahan[11], David A. Hafler[12], Ruth R. Montgomery[12], Albert C. Shaw[12], Steven H. Kleinstein [12], Jeremy P. Gygi[12], Dylan Duchen[12], Shrikant Pawar[12], Anna Konstorum[12], Ernie Chen[12], Chris Cotsapas[12], Xiaomei Wang[12], Charles Dela Cruz[12], Akiko Iwasaki[12], Subhasis Mohanty[12], Allison Nelson[12], Yujiao Zhao[12], Shelli Farhadian[12], Hiromitsu Asashima[12], Omkar Chaudhary[12], Andreas Coppi[12], John Fournier[12], M. Catherine Muenker[12], Khadir Raddassi[12], Michael Rainone[12], William Ruff[12], Syim Salahuddin[12], Wade L. Shulz[12], Pavithra Vijayakumar[12], Haowei Wang[12], Esio Wunder Jr.[12], H. Patrick Young[12], Albert I. Ko[12], Gisela Gabernet[12], Denise Esserman[12], Leying Guan[12], Anderson Brito[12], Jessica Rothman[12], Nathan D. Grubaugh[12], Kexin Wang[12], Leqi Xu[12], Holden Maecker[13], Bali Pulendran[13], Kari C. Nadeau[13], Yael Rosenberg-Hasson[13], Michael Leipold[13], Natalia Sigal[13], Angela Rogers[13], Andrea Fernandes[13], Monali Manohar[13], Evan Do[13], Iris Chang[13], Alexandra S. Lee[13], Catherine Blish[13], Henna Naz Din[13], Jonasel Roque[13], Linda N. Geng[13], Maja Artandi[13], Mark M. Davis[13], Neera Ahuja[13], Samuel S. Yang[13], Sharon Chinthrajah[13], Thomas Hagan[13], Tyson H. Holmes[13], Koji Abe[13], Lindsey R. Baden[15,16], Kevin Mendez[15], Jessica Lasky-Su[15], Alexandra Tong[15], Rebecca Rooks[15], Michael Desjardins[15], Amy C. Sherman[15], Stephen R. Walsh[15], Xhoi Mitre[15], Jessica Cauley[15], Xiaofang Li[15], Bethany Evans[15], Christina Montesano[15], Jose Humberto Licona[15], Jonathan Krauss[15], Nicholas C. Issa[15], Jun Bai Park Chang[15], Natalie Izaguirre[15], David B. Corry[18], Farrah Kheradmand[18], Li-Zhen Song[18], Ebony Nelson[18], Monica Kraft[19], Chris Bime[19], Jarrod Mosier[19], Heidi Erickson[19], Ron Schunk[19], Hiroki Kimura[19], Michelle Conway[19], Dave Francisco[19], Allyson Molzahn[19], Connie Cathleen Wilson[19], Ron Schunk[19], Trina Hughes[19], Bianca Sierra[19], Jordan Oberhaus[20], Faheem W. Guirgis[20], Brittney Borresen[21], Matthew L. Anderson[21], Bjoern Peters[22], James A. Overton[22], Randi Vita[22], Kerstin Westendorf[22], Scott J. Tebbutt[23], Casey P. Shannon[23], Rafick-Pierre Sekaly[24], Slim Fourati[24], Grace A. McComsey[24], Paul Harris[24], Scott Sieg[24], George Yendewa[24], Mary Consolo[24], Heather Tribout[24], Susan Pereira Ribeiro[24], Charles B. Cairns[25], Elias K. Haddad[25], Michele A. Kutzler[25], Mariana Bernui[25], Gina Cusimano[25], Jennifer Connors[25], Kyra Woloszczuk[25], David Joyner[25], Carolyn Edwards[25], Edward Lee[25], Edward Lin[25], Nataliya Melnyk[25], Debra L. Powell[25], James N. Kim[25], I. Michael Goonewardene[25], Brent Simmons[25], Cecilia M. Smith[25], Mark Martens[25], Brett Croen[25], Nicholas C. Semenza[25], Mathew R. Bell[25], Sara Furukawa[25], Renee McLin[25], George P. Tegos[25], Brandon Rogowski[25], Nathan Mege[25], Kristen Ulring[25], Pam Schearer[25], Judie Sheidy[25], Crystal Nagle[25], James A. Overton[26], Scott R. Hutton[27], Greg Michelotti[27], Kari Wong[27], Adeeb Rahman[28] & Vicki Seyfert-Margolis[29]

[18]Baylor College of Medicine and the Center for Translational Research on Inflammatory Diseases, Houston, TX, USA. [19]University of Arizona, Tucson, AZ, USA. [20]University of Florida, Jacksonville, FL, USA. [21]University of South Florida, Tampa, FL, USA. [22]La Jolla Institute for Immunology, La Jolla, CA, USA. [23]Prevention of Organ Failure (PROOF) Centre of Excellence, University of British Columbia, Vancouver, BC, Canada. [24]Case Western Reserve University and University Hospitals of Cleveland, Cleveland, OH, USA. [25]Drexel University, Tower Health Hospital, Philadelphia, PA, USA. [26]Knocean Inc, Toronto, ON, Canada. [27]Metabolon Inc, Morrisville, NC, USA. [28]Immunai Inc, New York, NY, USA. [29]MyOwnMed Inc., Bethesda, MD, USA.

