## [Transparent Peer Review file · Nature Communications]

Host-Microbe Multiomic Profiling Identifies Distinct COVID-19 Immune Dysregulation in Solid Organ Transplant Recipients

Corresponding Author: Dr Charles Langelier

Version 0:

Reviewer comments:

Reviewer #1

(Remarks to the Author)

In this study, Pickering et al., evaluated, among COVID-19 patients, the PBMC and nasal transcriptional profiling of 86 solid organ transplant (SOT) compared to 172 non-SOT controls. Their data shows interesting patterns of upregulation of innate immune pathways related to interferon and Toll-like receptor signaling, and complement activation, among SOT recipients. In parallel, microbiome alpha diversity was higher in SOT recipients and they had impaired SARS-CoV-2 clearance. Mass cytometry of PBMCs showed decreased plasmablasts and transitional B cells, and increased senescent T cells. Severity of the disease among SOT was associated with a less robust induction of inflammatory chemokines, such as IL-6 and CCL7, and a more blunted proinflammatory transcriptional response in the blood and airway.

Overall the manuscript is well written, very clear and describes some provocative and important findings.

I have a few comments to make.

Below are my point-by-point comments

Major Comments

1. For the "Cytokine and chemokine expression upon hospitalization and over time" described in line 175, the authors adjusted their analyses by viral abundance noting that the differences identified between the SOT and no-SOT group did not change much for some. While that is an important analysis, it did not test for correlations between viral abundance and Cytokine/chemokine expression. That is a separate question but would be interesting to know.
2. Line 234: "despite the significant difference in the nasal viral rpM (Fig. 2a), no differentially expressed genes were identified between groups at a false discovery rate (FDR) < 0.05 at the time of hospital admission. GSEA nonetheless demonstrated that SOT recipients exhibited increased expression of genes related to IL-10 signaling, neutrophil degranulation, type I IFN signaling, IL-1, and IL-4/IL-13 signaling in the upper respiratory tract at the time of hospital admission." This does not make sense to me. If there are no differently enriched genes not sure what data went into GSEA to get "increased expression of genes related to IL-10 signaling, neutrophil degranulation, type I IFN signaling, IL-1, and IL-4/IL-13 signaling in the upper respiratory tract at the time of hospital admission". I get that under GSEA genes are pooled into signaling pathways and can show differences in pathways when individual genes may not reach FDR but the authors should clarify the approach, thresholds and codes used. This will be important to clarify all the other GSEA analyses performed.
3. In line to comment #1, line 240: "Most inflammatory pathways differentially upregulated in SOT recipients were unaffected by viral rpM adjustment". Where there nasal inflammatory pathways that correlated with nasal viral abundance?
4. Regarding the Airway microbiome differences between SOT recipients and controls, I do not see any description in the results about taxonomic differences shown in Panel F of Figure 6.
5. Line 359 in the discussion: "while ISG expression in the blood strongly correlated with viral rpM in non-SOT controls, this relationship was not consistently observed in SOT recipients, suggesting a partial decoupling of IFN signaling and viral pathogen burden". "Decoupling" is a descriptive term. How do the authors interpret this finding, what are the possible explanations or significance of this finding?
6. Limitation paragraph should include lack of data on what's going on in the lower airways, which is the primary site for disease and what ultimately leads to the mortality of these patients.

Minor Comments

1. Please clarify that antibodies were only measured in blood and not in nasal samples. This should be added to the methods (line 502) and in the text (line 161)
2. Analytical codes cannot be found in https://bitbucket.org/kleinstein/impacc-public-code/src/master/SOT_manuscript.

(Remarks on code availability)

codes are not available at the time I reviewed it

Reviewer #2

(Remarks to the Author)

In this study, Pickering and colleagues aimed to deconvolute the immune-response against SARS-CoV-2 at distinct biological levels using different high-throughput technologies, in SOT hospitalized for COVID-19. They took advantage of a unique prospectively collected samples in a longitudinal cohort study (IMPACC) from roughly 20 hospitals in the US, with the goal to decipher the contributions of the pathogen and host immune response in modulating the manifestations, severity, and course of COVID-19 and to identify potential biomarkers as well as inform therapeutic interventions.

They thoroughly assess at different biological levels, circulating immune cell phenotypes using mass cytometry, serum anti-SARS-CoV-2 IgG Ab and chemokine profiles, and peripheral as well as nasal swab transcriptional profiling and microbiome in SOT (mainly kidney transplants, 52%) as compared to clinically matched non-SOT individuals. They compared these biological features between SOT and non-SOT and regarding the capacity of viral clearance after acute infection, between distinct clinical trajectory groups stratified according to infection severity (T1-3 mild to moderate, T4 long hospitalization and prolonged respiratory support requirement, T5 death within the following 28 days) and at distinct time points during convalescence.

Overall, they show that as compared to non-SOT, SOT display increased viral abundance/replication with an impaired viral clearance, lower plasmablasts and transitional B cells but higher proportions of CD4+ and CD8+ EMRA T cells, with lower IgG levels early after infection but reaching similar titers than non-SOT over time. Also, some differences were observed on total chemokine levels, with lower IFN- γ , OSM, and TNSF14 in SOT than non-SOT, also when adjusted by viral rprm, and a CXCL11, CCL3/4 rise over time in SOT but not in non-SOT. Higher up-regulation of innate immune related pathways and maintenance of higher levels over time of types II/III IFN and costimulation was especially observed among SOT.

Surprisingly, no major transcriptional differences in nasal swabs were observed between groups, despite difference on nasal viral rpM at admission. Longitudinal assessment over time seems to mirror transcriptional changes in the blood with increased expression of genes related to IFN among SOT. A certain decoupling between type I IFN and rpM in blood and airways was observed, with lower correlation in blood of SOT but not in the upper respiratory tract. On the other hand, the study of airway Microbiome metatranscriptomics showed higher alpha diversity in SOT than non-SOT, whereas beta diversity did not differ between groups.

Authors then correlate all these biological findings in SOT and non-SOT with different infection trajectories/severities (T1-3 vs T4-5). Here, both SOT and non-SOT displayed similarly reduced cell subsets, with some few specific differences, severe non-SOT displayed higher proinflammatory serum cytokines and chemokines, whereas SOT lower IFN- γ and IL21B, both severe SOT and non-SOT exhibited greater expression of different immune pathways including neutrophil degranulation, innate immune system signaling, and cellular responses to stress, with a unique downregulation in SOT of genes related to TCR and CD28 signaling, and TNF receptor-ligand interactions. In the upper airways, differently to non-SOT, severe SOT showed downregulation of TLR signaling genes as well as neutrophil degranulation and innate immune signaling.

Comments.

Interesting work analyzing multiple immune parameters using diverse high-throughput technologies in a large multicenter SOT and non-SOT patient cohort. Authors should be acknowledged for the high amount of work done implementing such highly refined technologies and also to obtain such a huge and important sample biobank during the difficult times during the first wave of this pandemic.

The added value of this work is the combination of immune analysis done in naïve COVID-19 SOT infected during the first wave of the pandemic. The results of this work are valuable and provide granularity to the immune response triggered by SOT at the time of COVID19. Furthermore, authors have been able to assess both PB as well as the target respiratory tract to analyze biological differences, which further insight on the immune response during COVID19. Globally, the message is, however, not really new and is in line with previous publications, specifically dealing with SOT, showing similar findings but assessed individually at specific immune compartments with different immune platforms. It is surprising the absence of any citation or discussion with previous publications showing similar data (e.g. Sattler a JCI 2020, Sun Z JASN 2022; Favà A. AJT 2021, Favà A. Kidney Int 2022).

Specific questions

Where all the SOT of this study all those recruited in the IMPACC study? if not, what were the reasons for their inclusion, sample viability, proportion of patients in the different clinical trajectories? This could have biased the results of the study. Number of time points analyzed in each patient for each immunological test should be better depicted. There is no information in the methods section. Please describe how many patients were evaluated per time point and per immune assay. According to the figures, it seems that many less number of patients were analyzed after visit 1 for all biomarkers, so I wonder how robust are the analyzes of the changes of immune profiles after visit 1.

A main limitation of the study, as the authors state, is the lack of a more in-depth description of the SOT group, besides the types of SOT, which I fully agree that should not be considered as a limitation but a strength, but rather, the lack of information of the types of IS, time of COVID19 after transplant surgery, two major variables clearly associated to distinct clinical outcomes and immune responses at the time of (any) viral infection. Are there differences on the distinct immune profiles according to these variables? It seems obvious that the lower proinflammatory profile observed between SOT and non-SOT is driven by the immunosuppressive context of SOT, especially when comparing severe and mild COVID19 patients.

While downregulation of TCR signaling, CD28 costimulation seems to be explained by the type of current

immunosuppressants, mainly targeting T cells, the lower proportion of B cells and PB among SOT is difficult to explain. Can authors provide an alternative explanation?

I would perhaps tone down the idea of the similar hard outcomes related to COVID19 infection (death) between SOT and non-SOT, regardless the propensity-score reports cited by authors. Many other important epidemiological studies have showed the higher mortality rates of SOT and other chronically immunosuppressed patients.

(Remarks on code availability)

Version 1:

Reviewer comments:

Reviewer #1

(Remarks to the Author)

The authors have addressed my comments. Thanks

(Remarks on code availability)

I saw the codes there but did not assess their reproducibility

Reviewer #2

(Remarks to the Author)

Thank you for answering to all my questions.

While the data reported on Visit 1 is homogenous and numerous in terms of sample size, I am now somewhat disappointed to see the drastic reduction in patients/samples analyzed in the "over time" analysis, as it may significantly limit the power of the paired analysis. In other words, the follow-up analysis, if paired, is reduced down to 7-11 SOT if analyzed until visits 5 or 6, or to roughly 25 until visit 3. So I wonder how consistent are these data.

I think it would be worth to show the number of tests analyzed in the X axis, under each timepoint, so that the reader may understand this smaller patient samples analyzed over time. I would also highlight this issue in the limitations of the study.

While CNI and especially MMF have shown some inhibitory effects on B cells, as authors have literally described, I am still surprised on this reduction, especially of plasma blasts.

(Remarks on code availability)

Reviewer #3

(Remarks to the Author)

I echo the critique raised by reviewer 2 about the significant dropout rate of sample size (shown in Figure S11) might influence the conclusion. I am very surprised that the author didn't justify why the number of patients significantly dropped after even the first visit from the clinician's point of view. I am not sure if the numbers included escalation samples (mentioned in the "Common statistical analyses framework") or not. If not, this will make me more worried. Here are my major comments about the statistical methodologies used in the paper:

1. Please read the paper "q2-longitudinal: Longitudinal and Paired-Sample Analyses of Microbiome Data" and rerun the analyses for metagenomics data. Q2-longitudinal is tailored for microbiome data and is more advanced. Check if the conclusion is consistent.
2. Power and sample size calculations are needed to demonstrate the model has sufficient power when the number of patients significantly dropped after even the first visit.
3. The authors need to justify why all analyses exclude the escalation samples and give a rigorous definition of what escalation means. If the numbers in Figure S11 include the filtered escalation samples, then please provide another figure without the escalation samples.
4. The statistical analyses are not consistent. Why did only the visit 1 analysis take the age, sex, and baseline respiratory severity into account, but not the longitudinal analyses? Covariate adjustment is very important and needs to be applied to all analyses. Furthermore, all covariates listed in Table 1 should be included in the analyses, as the data seems very complete.
5. The notation system in the "Materials and Methods" is a disaster. I don't think it meets the publication quality of Nature Communications. My suggestion is to use clearer, simpler language to define every type of dataset and parameters in the linear, linear mixed, and generalized additive models. It can be particularly helpful for statisticians and non-statisticians to understand the analysis. For example, the response and predictor variables (including the interaction terms) should be clearly defined; the p-values corresponding to which coefficient need to be specified; any derived metrics such as alpha-diversity, and beta-diversity should be explicitly given; the null and alternative hypotheses should be written using mathematical language.
6. The data preprocessing for RNA-seq and metagenomics data is too naïve. Most of biological sequence count data suffers

from zero-inflation and over-dispersion, the choice of a simpler log normalization or quantile normalization is too arbitrary. Sensitivity analysis needs to be conducted to avoid false discoveries.

7. The use of generalized additive models (GAMs) is encouraged because it is more flexible than the linear model. However, I think GAMs should be applied to all analyses, not only For longitudinal analysis of SARS-CoV-2 nasal viral rpM and serum anti-Spike IgG.

8. Benjamini-Hochberg correction is for multiple testing. If there is only one test (e.g. Figure 4b, CX3CL1, or IFNG), I don't understand why the correction is needed. The authors need to check the same issue throughout the paper.

9. A better way to visualize Figure S11 is to show the number of tests analyzed in the X axis, under each visit, so that the reader may understand this smaller patient samples analyzed over time and be cautious of the paper's conclusion. The authors also need to emphasize this issue in the limitations of the study.

(Remarks on code availability)

Version 2:

Reviewer comments:

Reviewer #2

(Remarks to the Author)

The responses given regarding the number of samples assessed per patient at each specific time point are somewhat vague, and even though the authors argue that the longitudinal evaluation of the multiple biomarkers is a secondary outcome of this work, in the overall message of the study, this is not at all what one captures when reading the paper and is actually emphasized in both the introduction and discussion sections.

It is obvious that in any longitudinal analysis patients may be lost of follow-up for many reasons, including death (especially in severe diseases such as this one), but this is something that should be taken into account and most importantly described in detail, which is still not the case in the manuscript. Indeed, if most severely ill patients and those that died were not measured for the longitudinal analysis, then the results may be misleading and biased. Actually the curves displayed could significantly vary.

It seems that authors forgot or do not want to show the number of tests assessed per time point in the figures of the manuscript. The table provided in the response to the reviewer just highlights what this reviewer described in the previous revision, this should be shown in the paper as usually done. The total number of samples analysed in X axis, although important to show the effort done by the authors, it unfortunately does not provide any biological relevance in terms of the consistency on the paired analysis done.

If authors aim to only describe the differences on the different biomarkers and immune responses between SOT and CTL at the time of hospitalization, which are the most consistent results, this should be much more emphasized and perhaps the paper should be redesigned.

(Remarks on code availability)

Reviewer #3

(Remarks to the Author)

The authors have thoroughly and thoughtfully addressed all of my concerns. Their revisions provide clear and well-substantiated responses, significantly improving the clarity and rigor of the manuscript. I am satisfied with the careful consideration they have given to each point raised, and I believe the manuscript is now much stronger as a result.

(Remarks on code availability)

Version 3:

Reviewer comments:

Reviewer #2

(Remarks to the Author)

Thanks for responding my questions. I am fine with the answers and changes done to the last version of the manuscript

(Remarks on code availability)

DEADLINE: May 6th, 2024
REVIEWER COMMENTS

Reviewer #1 (Remarks to the Author):

In this study, Pickering et al., evaluated, among COVID-19 patients, the PBMC and nasal transcriptional profiling of 86 solid organ transplant (SOT) compared to 172 non-SOT controls. Their data shows interesting patterns of upregulation of innate immune pathways related to interferon and Toll-like receptor signaling, and complement activation, among SOT recipients. In parallel, microbiome alpha diversity was higher in SOT recipients and they had impaired SARS-CoV-2 clearance. Mass cytometry of PBMCs showed decreased plasmablasts and transitional B cells, and increased senescent T cells. Severity of the disease among SOT was associated with a less robust induction of inflammatory chemokines, such as IL-6 and CCL7, and a more blunted proinflammatory transcriptional response in the blood and airway. Overall the manuscript is well written, very clear and describes some provocative and important findings.

I have a few comments to make.

Below are my point-by-point comments

Major Comments

1. For the Cytokine and chemokine expression upon hospitalization and over time described in line 175, the authors adjusted their analyses by viral abundance noting that the differences identified between the SOT and no-SOT group did not change much for some. While that is an important analysis, it did not test for correlations between viral abundance and Cytokine/chemokine expression. That is a separate question but would be interesting to know.

Response: We agree with the reviewer that the correlation between protein expression and SARS-CoV-2 viral abundance is a different but important and interesting question. Following the reviewer's suggestion and our analysis of interferon stimulated genes versus viral abundance (Supp. Fig. 8), we've added a new analysis assessing the correlation between protein expression and viral rpM (new Supp. Fig. 5b, c, see below). We calculated the linear regression slope between these two variables separately in the non-SOT and SOT groups, and found that CXCL8 was significantly associated with viral rpM in the SOT group, but not in the control group (Supp Fig. 5b). More specifically, we found that, in the control group, CXCL8 does not change with viral rpM, but in the SOT group, CXCL8 is positively associated with viral rpM (Supp. Figure 5c). We also now discuss this in the text as follows:

Line 189: We further analyzed the relationship between SARS-CoV-2 rpM and protein expression in each of the two study groups (Supp. Fig. 5b), and found a positive correlation between viral rpM and CXCL8 in the SOT recipients but not in controls (Supp. Fig. 5c).

Supplementary Figure 5. Effects of SARS-CoV-2 viral abundance on differential protein expression at Visit 1, and on the dynamics of immune proteins. (a) Bar plots showing serum proteins that are differentially expressed between SOT recipients and non-SOT controls at Visit 1, with (right) and without (left) controlling for SARS-CoV-2 viral rpM. P-values were calculated using a linear modeling and Benjamini-Hochberg correction. (b) Plot showing the slopes of serum protein expression against SARS-CoV-2 rpM. The x-axes and y-axes show the slopes in control and SOT groups, respectively. The black diagonal lines indicate $y = x$. (c) Plots showing CXCL8 protein expression versus viral rpM in control and SOT groups. In (b, c), P-values were calculated with linear model and Benjamini-Hochberg correction. (d) Scatter plot showing the dynamics of CXCL11, CCL3, CCL4 and IL22RA1 protein levels after hospital admission (adjusting for SARS-CoV-2 viral rpM). P-values were calculated using a linear mixed effects model and Benjamini-Hochberg correction.

2. Line 234: despite the significant difference in the nasal viral rpM (Fig. 2a), no differentially expressed genes were identified between groups at a false discovery rate (FDR) < 0.05 at the time of hospital admission. nonetheless demonstrated that SOT recipients exhibited increased expression of genes related to IL-10 signaling, neutrophil degranulation, type I IFN signaling, IL-1, and IL-4/IL-13 signaling in the upper respiratory tract at the time of hospital admission. This does not make sense to me. If there are no differently enriched genes not sure what data went into GSEA to get increased expression of genes related to IL-10 signaling, neutrophil degranulation, type I IFN signaling, IL-1, and IL-4/IL-13 signaling in the upper respiratory tract at the time of hospital admission. I get that under GSEA genes are pooled into signaling pathways and can show differences in pathways when individual genes may not reach FDR but the authors should clarify the approach, thresholds and codes used. This will be important to clarify all the other GSEA analyses performed.

Response: We would like to clarify that GSEA was originally designed to be used on all genes from a differential gene expression analysis, ranked by either log fold change or P-value (Subramanian, Tamayo, et al. (2005), PNAS 102, 15545-15550, <http://www.broad.mit.edu/gsea/>). As such, even genes that do not meet an adjusted P-value threshold for significance contribute to pathway enrichment analysis simply based on their ranking. We have now clarified this in the methods as follows:

Line 548: Fold-change values from all genes (regardless of their adjusted P-values) in the Visit 1 differential expression analyses, representing the fold-change of transplant patients over control patients, and longitudinal analyses, representing the interaction term of days post-admission and transplant status, were used as the input for Gene Set Enrichment Analysis (GSEA).

3. In line to comment #1, line 240: Most inflammatory pathways differentially upregulated in SOT recipients were unaffected by viral rpM adjustment. Where there nasal inflammatory pathways that correlated with nasal viral abundance?

Response: We thank the reviewer for this comment and now include a new Supplementary Figure 9 (see below) that highlights the nasal inflammatory pathways correlated with viral abundance, which included pathways related to type I and type II interferon signaling, TLR signaling and neutrophil degranulation, and were generally similar between SOT recipients and controls.

Supplementary Figure 9. Nasal biological signaling pathways associated with SARS-CoV-2 viral abundance. GSEA = gene set enrichment analysis. rpM = reads per million. Filled boxes indicate pathways demonstrating statistically significant enrichment (adjusted $P < 0.05$). P-values were calculated using a linear modeling and Benjamini-Hochberg correction.

4. Regarding the Airway microbiome differences between SOT recipients and controls, I do not see any description in the results about taxonomic differences shown in Panel F of Figure 6.

Response: We have now ensured that panel F is referenced, and thank the reviewer for highlighting this oversight.

Line 280: Finally, we asked whether specific taxa differed between groups, and found that only SARS-CoV-2 was significantly more abundant in the SOT recipients (Fig. 6f).

5. Line 359 in the discussion: while ISG expression in the blood strongly correlated with viral rpM in non-SOT controls, this relationship was not consistently observed in SOT recipients, suggesting a partial decoupling of IFN signaling and viral pathogen burden. Decoupling is a descriptive term. How do the authors interpret this finding, what are the possible explanations or significance of this finding?

Response: We interpret this finding as suggesting that extrinsic factors other than direct innate immune responses to viral nucleic acid are driving the augmented systemic interferon response observed in SOT recipients. This differs from results in the upper airway, where the relationship is preserved between viral load and ISG expression regardless of SOT status. This observation

may be the result of compensatory innate immune activation in SOT recipients who have impaired adaptive immune responses. We have now added the following to the text:

Line 372: This finding suggests that factors other than a direct response to viral nucleic acid may be driving the augmented systemic interferon signaling observed in SOT recipients. Perhaps this reflects non-specific compensatory innate immune activation in the setting of impaired adaptive immunity in SOT recipients.

6. Limitation paragraph should include lack of data on what's going on in the lower airways, which is the primary site for disease and what ultimately leads to the mortality of these patients.

Response: We have now added this important limitation to the discussion.

Line 419: ...a lack of data from the primary site of infection in the lower airway,...

Minor Comments

1. Please clarify that antibodies were only measured in blood and not in nasal samples. This should be added to the methods (line 502) and in the text (line 161)

Response: We have clarified that antibodies were indeed only measured in the blood and not in nasal samples as follows:

Line 519: Antibody levels against the recombinant SARS-CoV-2 spike protein receptor-binding domain (RBD) were measured in the blood using a research-grade enzyme-linked immunosorbent assay (ELISA) as described¹⁷.

2. Analytical codes cannot be found in https://bitbucket.org/kleinstein/impacc-public-code/src/master/SOT_manuscript.

Response: We apologize for the typo in the manuscript. The link has been corrected to: https://bitbucket.org/kleinstein/impacc-public-code/src/master/solid_organ_transplant_manuscript/

Reviewer #1 (Remarks on code availability):

codes are not available at the time I reviewed it

Response: We apologize for the typo in the manuscript. The link has been corrected to: https://bitbucket.org/kleinstein/impacc-public-code/src/master/solid_organ_transplant_manuscript/

Reviewer #2 (Remarks to the Author):

In this study, Pickering and colleagues aimed to deconvolute the immune-response against SARS-CoV-2 at distinct biological levels using different high-throughput technologies, in SOT hospitalized for COVID-19. They took advantage of a unique prospectively collected samples in a longitudinal cohort study (IMPACC) from roughly 20 hospitals in the US, with the goal to decipher the contributions of the pathogen and host immune response in modulating the manifestations, severity, and course of COVID-19 and to identify potential biomarkers as well as inform therapeutic interventions.

They thoroughly assess at different biological levels, circulating immune cell phenotypes using mass cytometry, serum anti-SARS-CoV-2 IgG Ab and chemokine profiles, and peripheral as well as nasal swab transcriptional profiling and microbiome in SOT (mainly kidney transplants, 52%) as compared to clinically matched non-SOT individuals. They compared these biological features between SOT and non-SOT and regarding the capacity of viral clearance after acute infection, between distinct clinical trajectory groups stratified according to infection severity (T1-3 mild to moderate, T4 long hospitalization and prolonged respiratory support requirement, T5 death within the following 28 days) and at distinct time points during convalescence.

Overall, they show that as compared to non-SOT, SOT display increased viral abundance/replication with an impaired viral clearance, lower plasmablasts and transitional B cells but higher proportions of CD4+ and CD8+ EMRA T cells, with lower IgG levels early after infection but reaching similar titers than non-SOT over time. Also, some differences were observed on total chemokine levels, with lower IFN- γ , OSM, and TNF14 in SOT than non-SOT, also when adjusted by viral rprm, and a CXCL11, CCL3/4 rise over time in SOT but not in non-SOT. Higher up-regulation of innate immune related pathways and maintenance of higher levels over time of types II/III IFN and costimulation was especially observed among SOT. Surprisingly, no major transcriptional differences in nasal swabs were observed between groups, despite difference on nasal viral rpM at admission. Longitudinal assessment over time seems to mirror transcriptional changes in the blood with increased expression of genes related to IFN among SOT. A certain decoupling between type I IFN and rpM in blood and airways was observed, with lower correlation in blood of SOT but not in the upper respiratory tract. On the other hand, the study of airway Microbiome metatranscriptomics showed higher alpha diversity in SOT than non-SOT, whereas beta diversity did not differ between groups.

Authors then correlate all these biological findings in SOT and non-SOT with different infection trajectories/severities (T1-3 vs T4-5). Here, both SOT and non-SOT displayed similarly reduced cell subsets, with some few specific differences, severe non-SOT displayed higher proinflammatory serum cytokines and chemokines, whereas SOT lower IFN- γ and IL21B, both severe SOT and non-SOT exhibited greater expression of different immune pathways including neutrophil degranulation, innate immune system signaling, and cellular responses to stress, with a unique downregulation in SOT of genes related to TCR and CD28 signaling, and TNF receptor-ligand interactions. In the upper airways, differently to non-SOT, severe SOT showed downregulation of TLR signaling genes as well as neutrophil degranulation and innate immune signaling.

Comments.

Interesting work analyzing multiple immune parameters using diverse high-throughput technologies in a large multicenter SOT and non-SOT patient cohort. Authors should be acknowledged for the high amount of work done implementing such highly refined technologies

and also to obtain such a huge and important sample biobank during the difficult times during the first wave of this pandemic.

The added value of this work is the combination of immune analysis done in naïve COVID-19 SOT infected during the first wave of the pandemic. The results of this work are valuable and provide granularity to the immune response triggered by SOT at the time of COVID19. Furthermore, authors have been able to assess both PB as well as the target respiratory tract to analyze biological differences, which further insight on the immune response during COVID19. Globally, the message is, however, not really new and is in line with previous publications, specifically dealing with SOT, showing similar findings but assessed individually at specific immune compartments with different immune platforms. It is surprising the absence of any citation or discussion with previous publications showing similar data (e.g. Sattler a JCI 2020, Sun Z JASN 2022; Favà A. AJT 2021 Favà A. Kidney Int 2022).

Response: We appreciate this input and have now cited these important publications.

Line 305: In both SOT recipients and controls, severe disease was characterized by reductions in several immune cell populations, including conventional dendritic cells (DCs), intermediate (CD14+CD16+) monocytes, and several CD4+ T cell subsets, as has been previously observed^{17,21} (Fig. 7a).

Line 81: Few studies, however, have profiled the immune landscape of SOT recipients in the context of severe infection¹²⁻¹⁴,...

Specific questions

Where all the SOT of this study all those recruited in the IMPACC study? if not, what were the reasons for their inclusion, sample viability, proportion of patients in the different clinical trajectories? This could have biased the results of the study.

Response: We appreciate this question and would like to clarify that all SOT recipients were all those recruited by the IMPACC study.

Line 105: We conducted a case-control study of patients hospitalized for COVID-19 within the IMPACC cohort, which comprised 1164 patients enrolled across the US¹⁷⁻¹⁹ between May 2020 and March 2021. 86 SOT recipients from 11 medical centers were matched 2:1 by age, sex, and study site with 172 non-SOT controls from the same cohort (Fig. 1, Table 1).

Number of time points analyzed in each patient for each immunological test should be better depicted. There is no information in the methods section. Please describe how many patients were evaluated per time point and per immune assay. According to the figures, it seems that many less number of patients were analyzed after visit 1 for all biomarkers, so I wonder how robust are the analyzes of the changes of immune profiles after visit 1.

Response: We appreciate this suggestion and have now included a schematic of how many patients were evaluated per time point and per immune assay as a new Supplementary Figure 11:

Assay	Serum Olink	80	54	34	22	11	8
		162	97	72	47	25	21
	Serum Antibodies	76	51	34	21	11	7
		153	92	67	42	21	21
	PBMC Transcriptomics	66	50	31	20	12	7
		147	90	66	39	24	21
	Nasal Transcriptomics	63	39	27	19	7	7
		125	81	57	39	20	16
	Nasal Metatranscriptomics	69	42	28	19	8	7
		137	84	61	41	20	17
	Blood CyTOF	54	36	21	18	8	6
		107	74	47	34	19	17
		Visit 1	Visit 2	Visit 3	Visit 4	Visit 5	Visit 6

Supplementary Figure 11. Samples collected at each study visit for SOT recipients and non-SOT controls. SOT recipients are colored gold and controls colored blue. Bar graph X axis relates to the proportion of patients in each group with available samples.

A main limitation of the study, as the authors state, is the lack of a more in-depth description of the SOT group, besides the types of SOT, which I fully agree that should not be considered as a limitation but a strength, but rather, the lack of information of the types of IS, time of COVID19 after transplant surgery, two major variables clearly associated to distinct clinical outcomes and immune responses at the time of (any) viral infection.

Are there differences on the distinct immune profiles according to these variables? It seems obvious that the lower proinflammatory profile observed between SOT and non-SOT is driven by the immunosuppressive context of SOT, especially when comparing severe and mild COVID19 patients.

Response: We appreciate this comment and the reviewer's questions. Unfortunately, the timing of COVID-19 following transplant surgery was not captured in the study database. For the other important information on the SOT group, we have now included a new Supplementary Table 2 (see below) describing the immunosuppressants administered to SOT recipients in the cohort at the time of admission, which emphasizes the significant heterogeneity across patients in terms of medications, and the challenges in clearly discerning whether a specific regimen influenced immune signaling our outcomes. We refer to this in the manuscript as follows:

Line 110: Immunosuppressive regimens being taken at the time of hospital admission varied across SOT recipients, although mycophenolate and tacrolimus were the most common (Supp. Table 2).

Supplementary Table 2. Immunosuppressive treatment in SOT recipients at the time of hospital admission.

Immunosuppression	Heart¹	Kidney	Liver²	Lung	All
None reported	1	3	1	3	8
Azathioprine	0	0	1	0	1
Azathioprine, tacrolimus	0	1	1	0	2
Belatacept, mycophenolate	0	2	0	0	2
Cyclosporine	0	2	0	0	2
Cyclosporine, mycophenolate	0	1	0	0	1
Everolimus, tacrolimus	0	0	0	2	2
Mycophenolate	1	2	1	3	7
Mycophenolate, prednisone, tacrolimus	0	1	0	0	1
Mycophenolate, tacrolimus	6	17	7	7	37
Prednisone	0	1	1	0	2
Sirolimus, tacrolimus	1	0	0	0	1
Tacrolimus	2	11	5	2	20

¹1/11 heart transplant recipients had also received kidney transplants.

²3/17 liver transplant recipients had also received kidney transplants.

In addition, we performed three new analyses asking whether administration of either mycophenolate mofetil or tacrolimus, the two most widely used immunosuppressants, was associated with differences in SARS-CoV-2 abundance, spike IgG levels or disease severity. We found no significant differences. We have now added these results as a new Supplementary Figure 3 (see below) and describe them in the text as follows:

Line 117: Within the SOT group, we asked whether receipt of either mycophenolate or tacrolimus at the time of hospital admission influenced severity TG, but found no significant differences. Of patients receiving mycophenolate, 37.7% were in TG 4-5, compared with 18.8% of those not receiving mycophenolate (P=0.077). Of patients receiving tacrolimus, 30.4% were in TG 4-5, compared with 28.6% of those not receiving tacrolimus (P=0.88).

Line 143: ...or receipt of either mycophenolate or tacrolimus (Supp. Fig. 3a, b).

Line 168: Anti-SARS-CoV-2 spike IgG levels also did not differ based on receipt of either mycophenolate or tacrolimus (Supp. Fig. 3c, d).

Supplementary Figure 3. Relationship between immunosuppressant administration and viral abundance, SARS-CoV-2 IgG and COVID-19 severity. (a) Relationship between mycophenolate administration and fraction of patients with severe disease trajectories (IMPACC trajectory group 4 or 5). (b) Relationship between tacrolimus administration and fraction of patients with severe disease trajectories (IMPACC trajectory group 4 or 5). (c) Relationship between mycophenolate administration and SARS-CoV-2 rpM. (d) Relationship between tacrolimus administration and SARS-CoV-2 rpM. (e) Relationship between mycophenolate administration and SARS-CoV-2 spike IgG. (f) Relationship between tacrolimus administration and SARS-CoV-2 spike IgG. P-values were calculated by binomial logistic regression.

While downregulation of TCR signaling, CD28 costimulation seems to be explained by the type of current immunosuppressants, mainly targeting T cells, the lower proportion of B cells and PB among SOT is difficult to explain. Can authors provide an alternative explanation?

Response: We thank the reviewer for this question, and note that the two most common immunosuppressants administered to SOT recipients in the cohort were mycophenolate mofetil and tacrolimus, which inhibit both T and B cells proliferation. Mycophenolate leads to inhibition of inosine-5'-monophosphate dehydrogenase, thus depleting guanosine nucleotides in both T and B lymphocytes, inhibiting their proliferation. (PMID: 15803924). Tacrolimus binds FKBP12 blocking the activation of the cell cycle-related kinase TOR, which in turn inhibits cell-cycle progression. This ultimately inhibits antigen-induced proliferation of T and B cells, as well as antibody production. (PMID: 12742462).

I would perhaps tone down the idea of the similar hard outcomes related to COVID19 infection (death) between SOT and non-SOT, regardless the propensity-score reports cited by authors. Many other important epidemiological studies have showed the higher mortality rates of SOT and other chronically immuno suppressed patients.

Response: We appreciate this suggestion and have toned down the language related to this idea in the abstract, introduction and discussion.

Reviewer 2

Thank you for answering to all my questions.

While the data reported on Visit 1 is homogenous and numerous in terms of sample size, I am now somewhat disappointed to see the drastic reduction in patients/samples analyzed in the "over time" analysis, as it may significantly limit the power of the paired analysis. In other words, the follow-up analysis, if paired, it is reduced down to 7-11 SOT if analyzed until visits 5 or 6, or to roughly 25 until visit 3. So I wonder how consistent are these data.

I think it would be worth to show the number of tests analyzed in the X axis, under each timepoint, so that the reader may understand this smaller patient samples analyzed over time. I would also highlight this issue in the limitations of the study.

We appreciate this point, as well as the reviewer's suggestion, and agree that the decrease in sample size at later visits is a limitation of the study. That said, we would like to underscore that the central findings and conclusions from our study are primarily based on Visit 1 analyses. Our longitudinal transcriptomic and proteomic assessments for the most part provide complementary information to support and enrich the Visit 1 findings.

We would also like to note that in longitudinal cohort studies of hospitalized patients with COVID-19 and other acute illnesses, especially studies that seek to collect biological samples over 28 days, it is expected that the number of patients remaining in the study after 28 days, 14 days, or even 7 days, will be significantly lower than at earlier time points. Many patients recover from their acute illnesses and get discharged within 14 days or less; and others unfortunately die from their illness before 28 days has elapsed, something especially common during the early months of the COVID-19 pandemic. Consequently, these patients are no longer available to provide samples at later timepoints, and thus drop out of the study. This is a universal challenge for longitudinal observational cohort studies of hospitalized patients and in particular critically ill COVID-19 patients (e.g., Nature Communications, PMID: 34716338; Nature Communications, PMID: 38830853).

Therefore, the decrease in patient numbers at later time points is really quite expected, and simply reflects the natural course of hospitalization and patient outcomes, including recovery and discharge or in-hospital mortality. Nonetheless, we agree the decrease sample size at later visits is a limitation of our study, and have therefore added the following to the discussion:

Line 427: "Additionally, our sample size did decrease at timepoints further out from hospital admission, as is the case for most longitudinal observational studies of hospitalized patients. Therefore, we primarily focused on findings at Visit 1, and our longitudinal findings should be interpreted with this in mind."

In addition, we have updated Supp. Figure 11 as suggested to include the total number of tests at each visit.

Assay	Visit 1		Visit 2		Visit 3		Visit 4		Visit 5		Visit 6		Total	
	SOT	Ctl	SOT	Ctl	SOT	Ctl	SOT	Ctl	SOT	Ctl	SOT	Ctl	SOT	Ctl
Serum Olink	80	162	54	97	34	72	22	47	11	25	8	21	SOT: 209	Ctl: 424
	Total: 242		Total: 151		Total: 106		Total: 69		Total: 36		Total: 29		Total: 633	
Serum Antibodies	76	153	51	92	34	67	21	42	11	21	7	21	SOT: 200	Ctl: 396
	Total: 229		Total: 143		Total: 101		Total: 63		Total: 32		Total: 28		Total: 596	
PBMC Transcriptomics	66	147	50	90	31	66	20	39	12	24	7	21	SOT: 186	Ctl: 387
	Total: 213		Total: 140		Total: 97		Total: 59		Total: 36		Total: 28		Total: 573	
Nasal Transcriptomics	63	125	39	81	27	57	19	39	7	20	7	16	SOT: 162	Ctl: 338
	Total: 188		Total: 120		Total: 84		Total: 58		Total: 27		Total: 23		Total: 500	
Nasal Metatranscriptomics	69	137	42	84	28	61	19	41	8	20	7	17	SOT: 173	Ctl: 360
	Total: 206		Total: 126		Total: 89		Total: 60		Total: 28		Total: 24		Total: 533	
Blood CyTOF	54	107	36	74	21	47	18	34	8	19	6	17	SOT: 143	Ctl: 298
	Total: 161		Total: 110		Total: 68		Total: 52		Total: 27		Total: 23		Total: 441	

Supplementary Figure 11. Samples collected at each study visit for SOT recipients and non-SOT controls. SOT recipients are colored gold and controls colored blue. Bar graph X axis relates to the proportion of patients in each group with available samples.

While CNI and especially MMF have shown some inhibitory effects on B cells, as authors have literally described, I am still surprised on this reduction, especially of plasma blasts.

We were also initially surprised; however, a review of the literature does suggest that others have observed inhibition of plasmablasts by MMF (e.g., Arthritis Research & Therapy, PMID: 22571761; PMID: 25890338).

Reviewer 3

I echo the critique raised by reviewer 2 about the significant dropout rate of sample size (shown in Figure S11) might influence the conclusion. I am very surprised that the author didn't justify why the number of patients significantly dropped after even the first visit from the clinician's point of view. I am not sure if the numbers included escalation samples (mentioned in the "Common statistical analyses framework") or not. If not, this will make me more worried. Here are my major comments about the statistical methodologies used in the paper:

We appreciate these points and would like to again emphasize that the central findings and conclusions from our study are based on visit 1 analyses. Our longitudinal transcriptomic and proteomic assessments at days 0-3,4,7,14,21 and 28 provide additional context that complements the primary visit 1 findings.

As discussed in our response to reviewer 2, in longitudinal cohort studies of hospitalized patients with COVID-19, it is expected that the number of patients remaining in the study after 28 days, 14 days, or even 7 days, will be significantly lower than at earlier time points. Many

patients recover from their acute illnesses and get discharged within 14 days or less; and others may die from their illness before 28 days has elapsed, something especially common during the early months of the COVID-19 pandemic. Consequently, these patients are no longer available to provide samples at later timepoints, and thus drop out of the study. This is a universal challenge for longitudinal observational cohort studies of hospitalized patients and in particular critically ill COVID-19 patients (e.g., Nature Communications, PMID: 34716338; Nature Communications, PMID: 38830853).

Therefore, as noted above, the decrease in patient numbers at later time points is expected, and simply reflects the natural course of hospitalization and patient outcomes, including recovery and discharge or in-hospital mortality. Nonetheless, we agree the decrease sample size at later visits is a limitation of our study, and have therefore added the following to the discussion:

Line 427: “Additionally, our sample size did decrease at timepoints further out from hospital admission, as is the case for most longitudinal observational studies of hospitalized patients. Therefore, we primarily focused on findings at Visit 1, and our longitudinal findings should be interpreted with this in mind.”

1. Please read the paper “q2-longitudinal: Longitudinal and Paired-Sample Analyses of Microbiome Data” and rerun the analyses for metagenomics data. Q2-longitudinal is tailored for microbiome data and is more advanced. Check if the conclusion is consistent.

Thank you for suggesting this excellent paper. We have repeated the microbiome analysis analyzing metagenomic samples for transplant patients compared to the matched controls using the rigorous “Analysis of Compositions of Microbiomes with Bias Correction” (ANCOM-BC) tool, which was recently found to outperform other common statistical frameworks used for metagenomics (Nature Communications, PMID: 32665548). ANCOM-BC replicated our findings demonstrating that the only significantly different genus between groups was Betacoronavirus (p.adj = 0.0002). We now describe this additional validation analysis as follows:

Line 513: “Additionally, to confirm the finding from linear mixed effect modeling that Betacoronavirus was the only taxa with significant relative abundance changes in SOT recipients, we analyzed Visit 1 samples with “Analysis of Compositions of Microbiomes with Bias Corrections” (ANCOM-BC)²⁷ which also identified Betacoronavirus as the only significant differentially abundant taxon.”

2. Power and sample size calculations are needed to demonstrate the model has sufficient power when the number of patients significantly dropped after even the first visit.

In our study, calculations of power and sample sizes were not computed *a priori* due to the discovery nature of the parent IMPACC cohort and the lack of advance information on relevant transcriptional or proteomic parameters. The IMPACC cohort was conceived and began enrollment during the first few months of the COVID-19 pandemic, when neither biological nor clinical outcomes of COVID-19 were well understood. In our case, we analyzed every SOT recipient enrolled in one of the largest COVID-19 cohorts established to date. We then carried out a 2:1 matched case-control analysis.

Regarding post-hoc power calculations, we respectfully note multiple publications that demonstrate such calculations are not informative and in fact advise against their use (American Statistician, <https://doi.org/10.1198/000313001300339897>; Annals of Surgery, PMID: 29994928;

General Psychiatry, PMID: 31552383; Current Psychology, PMID: 32523323). Perhaps more importantly, however, is the fact that our analyses returned statistically significant results, which demonstrates that our study was sufficiently powered to identify significant differences between groups.

We nonetheless appreciate the importance of confirming the robustness and reproducibility of our findings. Permutation analysis is a well-established method (Statistical Sciences. 2004, <https://www.jstor.org/stable/4144438>) for testing the robustness of biological findings, and thus we have performed permutation analysis on key exemplary results from the longitudinal analyses. Using 1000 permutations of our variable of interest, solid organ transplant status, we found that:

- 1) SARS-CoV-2 nasal rpM remained significantly higher longitudinally in transplant recipients ($P = 0.0022$, permuted- $P = 0.006$)
- 2) CXCL11 remained significantly elevated longitudinally in transplant recipients ($P = 0.0042$, permuted- $P = 0.002$)
- 3) All differentially expressed genes identified in the PBMC longitudinal analysis, after FDR correction, remained statistically significant (permuted- $P < 0.001$)

In sum, we found that the significance of these key longitudinal results was confirmed by permutation analysis, supporting the robustness of the findings. We have now added these permutation analyses to the manuscript as follows:

Line 487: “Additionally, to confirm the robustness of key longitudinal analyses for viral load, Olink cytokines and PBMC genes, we performed permutation analysis²⁴ using 1000 iterations (randomly permuting the patient’s transplant/control group assignment 1000 times, and then comparing the observed test statistic to this distribution to assess its significance), and calculated the P-value as follows:

permuted- $P = (\text{number of iterations with test statistic more extreme than the observed test statistic}) / 1000$

For each of these three validation analyses, our findings remained statistically significant (permuted- $P < 0.01$).”

3. The authors need to justify why all analyses exclude the escalation samples and give a rigorous definition of what escalation means. If the numbers in Figure S11 include the filtered escalation samples, then please provide another figure without the escalation samples.

We appreciate the opportunity to clarify that ‘escalation samples’ in the IMPACC cohort refer to additional samples collected when a patient escalated from ward to ICU level care. We have now clarified this definition in the methods section as follows:

Line 475: “...and excluding eight additional samples (seven controls, one SOT recipient) collected when a patient was transferred from the ward to intensive care unit.”

We chose to exclude escalation samples because 1) there were only 8 escalation samples in the population studied, 7 of them were in controls and only 1 was in a SOT recipient, and 2) they were not collected at consistent timepoints, and doing so would bias towards patients with more severe disease having more samples included in analyses.

4. The statistical analyses are not consistent. Why did only the visit 1 analysis take the age, sex, and baseline respiratory severity into account, but not the longitudinal analyses? Covariate adjustment is very important and needs to be applied to all analyses. Furthermore, all covariates listed in Table 1 should be included in the analyses, as the data seems very complete.

We apologize for the confusion as we have realized that we inadvertently wrote that age, sex and respiratory severity were controlled for in Visit 1 analyses. Since our study design involved matching of cases (SOT recipients) and controls based on age, sex and enrollment site (as described in the Methods and Figure 1), it was not necessary to adjust for these covariates. We have corrected this error in the text.

Regarding the need to adjust for additional covariates: because no clinical outcomes (including respiratory severity) differed between cases and controls (Table 1), we did not adjust for differences in these variables, as doing so would be unnecessary and could lead to overfitting. We would like to note, however, that we did perform a dedicated, independent analysis focused exclusively on understanding immunological differences in severity/disease trajectory (Figure 7). Furthermore, we carried out our primary analyses with and without controlling for SARS-CoV-2 viral load, a variable which did in fact significantly differ based on SOT status (Supp. Figures 4, 5, 6 and 7). Finally, we also examined the relationship between viral load and gene expression in both PBMCs and the upper airway, in SOT cases and controls (Supp. Figure 8).

5. The notation system in the “Materials and Methods” is a disaster. I don’t think it meets the publication quality of Nature Communications. My suggestion is to use clearer, simpler language to define every type of dataset and parameters in the linear, linear mixed, and generalized additive models. It can be particularly helpful for statisticians and non-statisticians to understand the analysis. For example, the response and predictor variables (including the interaction terms) should be clearly defined; the p-values corresponding to which coefficient need to be specified; any derived metrics such as alpha-diversity, and beta-diversity should be explicitly given; the null and alternative hypotheses should be written using mathematical language.

We thank the reviewer for calling to our attention that our notations could be confusing to the broad audience of Nature Communications. We would like to clarify that we used the statistical programming language R to analyze the data. Thus, the notation system we used in the Materials and Methods section is standard R notation, and not mathematical notation. We chose to list the R formulae instead of mathematical formulae to help facilitate reproducibility of our analysis and to match with the publicly available code, as is increasingly standard practice (e.g., Nature Communications, PMID: 38902234; Nature Communications, PMID: 31201320; Nature Communications, PMID: 34795254; Nature Communications, PMID: 37857672; Nature Communications, PMID: 37996418).

More specifically, when we want to model a dependent variable based on a number of independent variables in R, we have to write the R formulae in our code. For example, the formula “ $z \sim x + y + x:y$ ” means that we’re modeling z as a function of x , y and the interaction term between x and y . We have clarified in the Materials and Methods section that the formulae written represent R formulae.

Furthermore, we have also clarified the independent variables, including the interaction terms (e.g., `event_date:transplant` is the R notation for the interaction term between `event_date` and `transplant` variables)

We have also provided the equations for alpha and beta diversity as requested:

Line 519: “Alpha diversity was calculated based on the Shannon Diversity Index:

$$H' = - \sum_{i=1}^s p_i \ln(p_i)$$

Where s is the number of species and p_i is the proportional abundance of species i .
Beta diversity was calculated based on the Bray-Curtis Dissimilarity Index:

$$BC_{jk} = 1 - \frac{\sum |x_{ij} - x_{ik}|}{\sum (x_{ij} + x_{ik})}$$

Where x_{ij} and x_{ik} represent the quantity of species (i) and two sites (j and k).”

6. The data preprocessing for RNA-seq and metagenomics data is too naïve. Most of biological sequence count data suffers from zero-inflation and over-dispersion, the choice of a simpler log normalization or quantile normalization is too arbitrary. Sensitivity analysis needs to be conducted to avoid false discoveries.

For bulk RNA-seq analyses we used the widely cited (>2,800 in PubMed), rigorous and conservative package, limma-voom (Genome Biology, PMID: 24485249). This included filtering of low abundance transcripts, calculation of normalization factors to scale library sizes, and quantile normalization. We have included below mean-variance plots pre- (A) and post- (B) filtering and normalization.

We have provided additional details in the methods to clarify this as follows:

Line 564: “We calculated normalization factors to scale library sizes using the calcNormFactors function from the edgeR package v3.40.2, then normalized the gene counts using the voom function (normalize.method = “quantile”) from the limma package v3.46.0, fitted a linear model for the gene expression with lmFit function (default settings), calculated the empirical Bayes statistics with eBayes function (default settings), and calculated the P values for differential expression controlling for FDR. We controlled for log-transformed viral rpM in certain analyses when indicated.”

For metagenomic analyses, it is a typical and accepted practice to normalize and evaluate taxa using relative abundance, and sometimes further normalize with a log transformation. For example, in the routinely used pipelines of the common tool qiime2 (highlighted in the paper suggested by the reviewer in comment #1; mSystems, PMID: 30505944), they too used the relative abundance of reads for taxa with linear mixed effect modeling. Compared to RNA-seq data, metagenomic data are in fact quite zero inflated, so we understand the reviewer's concerns despite this simplistic approach still being relatively common in the field. To address these concerns, we have repeated the analysis analyzing metagenomic samples for transplant patients compared to the matched controls using the rigorous "Analysis of Compositions of Microbiomes with Bias Correction" (ANCOM-BC) methodology which has been found to outperform other common statistical frameworks used for metagenomics (Nature Communications, PMID: 32665548). ANCOM-BC replicated our finding that the only significant differentially abundant genus was Betacoronavirus ($p_{adj} = 0.0002$).

7. The use of generalized additive models (GAMs) is encouraged because it is more flexible than the linear model. However, I think GAMs should be applied to all analyses, not only for longitudinal analysis of SARS-CoV-2 nasal viral rpM and serum anti-Spike IgG.

While we certainly recognize and appreciate the flexibility of GAM models, we purposely chose to use linear models for longitudinal analyses in this study to gain maximal insight into the biological relevance of differentially expressed genes in a manner not possible with GAM models. More specifically, linear models permitted assessing the overall directionality and rate of change in gene expression over the course of hospitalization for each patient. Slopes from linear regressions were used as input for gene set enrichment analysis (GSEA), which required a generalized estimation of gene expression trajectory by evaluating expression changes over time. This would be problematic with GAM models since one could only report the number of genes that changed over time without being able to interpret their global expression trajectory and associated biological relevance.

Because nasal viral rpM and anti-Spike IgG have been previously established as nonlinear (although both have been reasonably modeled with linear regressions as well), we opted to use more flexible GAM models for those two analyses.

Finally, as noted by the reviewer and discussed above in response to comment #1, our sample size decreases over time, as expected for longitudinal cohort studies of hospitalized patients studied for 28 days. GAM models require more frequent sampling to avoid overfitting (Journal of the Royal Statistical Society Series C: Applied Statistics, <https://doi.org/10.1111/rssc.12068>; Human Vaccines & Immunotherapy, PMID: 37697867) and thus would be more susceptible to errors from smaller sample sizes at later versus earlier time points.

8. Benjamini-Hochberg correction is for multiple testing. If there is only one test (e.g. Figure 4b, CX3CL1, or IFNG), I don't understand why the correction is needed. The authors need to check the same issue throughout the paper.

We would like to clarify that in Figure 4b, as in all proteomic analyses we performed, all 84 proteins that passed our QC filter (out of 92 proteins on the Olink proteomics panel) were analyzed. In Figure 4a, we highlight two exemplary differentially expressed proteins in the boxplots, and calculate P-values by adjusting for the complete multiplicity of testing. This is true throughout the manuscript when an adjusted P-value is reported.

9. A better way to visualize Figure S11 is to show the number of tests analyzed in the X axis,

under each visit, so that the reader may understand this smaller patient samples analyzed over time and be cautious of the paper’s conclusion. The authors also need to emphasize this issue in the limitations of the study.

We appreciate this suggestion and have updated Figure S11 as follows:

Assay	Sample Size						Total
	Visit 1	Visit 2	Visit 3	Visit 4	Visit 5	Visit 6	
Serum Olink	80	54	34	22	11	8	SOT: 209
	162	97	72	47	25	21	Ctl: 424
	Total: 242	Total: 151	Total: 106	Total: 69	Total: 36	Total: 29	Total: 633
Serum Antibodies	76	51	34	21	11	7	SOT: 200
	153	92	67	42	21	21	Ctl: 396
	Total: 229	Total: 143	Total: 101	Total: 63	Total: 32	Total: 28	Total: 596
PBMC Transcriptomics	66	50	31	20	12	7	SOT: 186
	147	90	66	39	24	21	Ctl: 387
	Total: 213	Total: 140	Total: 97	Total: 59	Total: 36	Total: 28	Total: 573
Nasal Transcriptomics	63	39	27	19	7	7	SOT: 162
	125	81	57	39	20	16	Ctl: 338
	Total: 188	Total: 120	Total: 84	Total: 58	Total: 27	Total: 23	Total: 500
Nasal Metatranscriptomics	69	42	28	19	8	7	SOT: 173
	137	84	61	41	20	17	Ctl: 360
	Total: 206	Total: 126	Total: 89	Total: 60	Total: 28	Total: 24	Total: 533
Blood CyTOF	54	36	21	18	8	6	SOT: 143
	107	74	47	34	19	17	Ctl: 298
	Total: 161	Total: 110	Total: 68	Total: 52	Total: 27	Total: 23	Total: 441

Supplementary Figure 11. Samples collected at each study visit for SOT recipients and non-SOT controls. SOT recipients are colored gold and controls colored blue. Bar graph X axis relates to the proportion of patients in each group with available samples.

We also agree the decrease sample size at later visits is a limitation of our study, and have added the following to the discussion, as note in our response to comment #1:

Line 427: “Additionally, our sample size did decrease at timepoints further out from hospital admission, as is the case for most longitudinal observational studies of hospitalized patients. Therefore, we primarily focused on findings at Visit 1, and our longitudinal findings should be interpreted with this in mind.”

Reviewer #2 (Remarks to the Author):

The responses given regarding the number of samples assessed per patient at each specific time point are somewhat vague, and even though the authors argue that the longitudinal evaluation of the multiple biomarkers is a secondary outcome of this work, in the overall message of the study, this is not at all what one captures when reading the paper and is actually emphasized in both the introduction and discussion sections.

It is obvious that in any longitudinal analysis patients may be lost of follow-up for many reasons, including death (especially in severe diseases such as this one), but this is something that should be taken into account and most importantly described in detail, which is still not the case in the manuscript. Indeed, if most severely ill patients and those that died were not measured for the longitudinal analysis, then the results may be misleading and biased. Actually the curves displayed could significantly vary.

It seems that authors forgot or do not want to show the number of tests assessed per time point in the figures of the manuscript. The table provided in the response to the reviewer just highlights what this reviewer described in the previous revision, this should be shown in the paper as usually done. The total number of samples analysed in X axis, although important to show the effort done by the authors, it unfortunately does not provide any biological relevance in terms of the consistency on the paired analysis done.

If authors aim to only describe the differences on the different biomarkers and immune responses between SOT and CTL at the time of hospitalization, which are the most consistent results, this should be much more emphasized and perhaps the paper should be redesigned.

We appreciate the reviewer's input, and believe we misinterpreted their specific request regarding how best to provide details on the number of samples for each assay at each timepoint. We apologize for this. We had attempted to address this request (as well as a similar one from Reviewer 3) with a comprehensive summary of the number of samples analyzed at each study visit for each assay, compiled into Supp. Figure 11. However, it is now clear that the request was to include this information into every figure containing longitudinal data, and so we have now done so. For example, please see figure 4, panel (c) below.

Figure 4. SOT recipients have higher levels of specific serum chemokines and lower levels of IFN-gamma. (a) Bar plots showing proteins that are differentially expressed between control and transplant patients at Visit 1 (adjusted $P < 0.05$). (b) Box plots showing the levels of CX3CL1 and IFNG at Visit 1. In (a, b), P-values were calculated using a linear model and Benjamini-Hochberg correction. (c) Scatter plot showing the dynamics of CXCL11 level after hospital admission (without adjusting for SARS-CoV-2 viral rPM). The ribbons indicate the 95% confidence interval of the linear mixed effects model fits. P-value was calculated using a linear mixed effects model and Benjamini-Hochberg correction. The number of patients sampled at each time point is depicted graphically below the X axis of panel (c).

To ensure complete transparency and provide further detail on every longitudinal data point included in figures, we have also now included a source data file with the manuscript that has the exact values for every datapoint displayed in the figures.

We've retained Supp. Figure 11 to further complement the data beneath each main manuscript figure related to a longitudinal analysis, as it provides comparable information but graphically depicts the number of samples at each study visit (as opposed to the number of samples analyzed within each time window following admission: 0-10 days, 10-20 days, 20-30 days post-admission).

With respect to the reviewer's concern regarding whether most severely ill patients and those that died were not measured in longitudinal analyses, we would like to respectfully note that only 5 SOT patients died (5.8% of the cohort), with a similar rate among the controls (4.7%). Thus, severely ill patients and those that died were well captured in the longitudinal analyses, and we have provided this information in table 1. We would also like to note that a similar rate of patients in the SOT group and controls were discharged by day 28 (80.8% SOT vs 83.1% controls, $P = 0.828$).

Lastly, we have de-emphasized the longitudinal aspect of the study in the introduction as suggested, and have revised the limitations section of the Discussion to include the following:

Line 429: “. In addition, our assessment of longitudinal trajectories was limited by a reduced number of patients still hospitalized at later timepoints in the study. Therefore, we primarily focused on findings at Visit 1, and our longitudinal findings should be interpreted with this in mind.”

Reviewer #3 (Remarks to the Author):

The authors have thoroughly and thoughtfully addressed all of my concerns. Their revisions provide clear and well-substantiated responses, significantly improving the clarity and rigor of the manuscript. I am satisfied with the careful consideration they have given to each point raised, and I believe the manuscript is now much stronger as a result.

We appreciate that the dedicated statistical reviewer agrees that we have thoroughly and thoughtfully addressed all of their concerns, and that our revisions have significantly improved the clarity and rigor of the manuscript.